# Meeting radiation dosimetry capacity requirements of population-scale exposures by geostatistical sampling

**Peter K. Rogan**[1,2]*, **Eliseos J. Mucaki**[1], **Ruipeng Lu**[3], **Ben C. Shirley**[2], **Edward Waller**[4], **Joan H. M. Knoll**[2,3]

**1** Department of Biochemistry, Schulich School of Medicine & Dentistry, University of Western Ontario, London, ON, Canada, **2** CytoGnomix Inc, London, ON, Canada, **3** Department of Pathology and Laboratory Medicine, Schulich School of Medicine & Dentistry, University of Western Ontario, London, ON, Canada, **4** Faculty of Energy Systems and Nuclear Science, OntarioTech University, Canada

* progan@uwo.ca

## Abstract

### Background

Accurate radiation dose estimates are critical for determining eligibility for therapies by timely triaging of exposed individuals after large-scale radiation events. However, the universal assessment of a large population subjected to a nuclear spill incident or detonation is not feasible. Even with high-throughput dosimetry analysis, test volumes far exceed the capacities of first responders to measure radiation exposures directly, or to acquire and process samples for follow-on biodosimetry testing.

### Aim

To significantly reduce data acquisition and processing requirements for triaging of treatment-eligible exposures in population-scale radiation incidents.

### Methods

Physical radiation plumes modelled nuclear detonation scenarios of simulated exposures at 22 US locations. Models assumed only location of the epicenter and historical, prevailing wind directions/speeds. The spatial boundaries of graduated radiation exposures were determined by targeted, multistep geostatistical analysis of small population samples. Initially, locations proximate to these sites were randomly sampled (generally 0.1% of population). Empirical Bayesian kriging established radiation dose contour levels circumscribing these sites. Densification of each plume identified critical locations for additional sampling. After repeated kriging and densification, overlapping grids between each pair of contours of successive plumes were compared based on their diagonal Bray-Curtis distances and root-mean-square deviations, which provided criteria (<10% difference) to discontinue sampling.

**Data Availability Statement:** Data and Software from this study are available in the Zenodo repository: https://doi.org/10.5281/zenodo.

3572574. The data includes modified U.S. state and sub-division boundary files [in KML format], as well as the geographic coordinates and dose values (World Geodetic System [WGS] 1984) which define each HPAC and geostatistical-derived plume for all scenarios described. The programs include custom scripts used to preprocess the data for use with ArcMap GIS software.

**Funding:** We are grateful to the SOSCIP Smart Computing for Innovation Consortium (J.H.M.K., P.K.R.), Natural Sciences and Engineering Research Council of Canada (Engage Program; E.W., P.K.R), Ontario Centres of Excellence (Talent Edge Postdoctoral Fellowship Program; J.H.M.K., P.K.R.), CytoGnomix (P.K.R.) for support of this project. We are grateful to Dr. Ruth C Wilkins (Health Canada) for valuable comments on this manuscript. The funders had no role in study design, data collection and analysis, decision to publish, or preparation of the manuscript.

**Competing interests:** Peter K. Rogan and Joan H. Knoll are founders and Ben Shirley is an employee of CytoGnomix, which partially supported this research. Cytognomix holds relevant patents and relevant patent applications. This does not alter our adherence to PLOS ONE policies on sharing data and materials.

## Results/Conclusions

We modeled 30 scenarios, including 22 urban/high-density and 2 rural/low-density scenarios under various weather conditions. Multiple (3–10) rounds of sampling and kriging were required for the dosimetry maps to converge, requiring between 58 and 347 samples for different scenarios. On average, 70±10% of locations where populations are expected to receive an exposure ≥2Gy were identified. Under sub-optimal sampling conditions, the number of iterations and samples were increased, and accuracy was reduced. Geostatistical mapping limits the number of required dose assessments, the time required, and radiation exposure to first responders. Geostatistical analysis will expedite triaging of acute radiation exposure in population-scale nuclear events.

## Introduction

In a large-scale nuclear event or accident, many will be exposed to varying levels of radiation, some of whom would require immediate treatment. Various approaches have been used to triage or estimate radiation exposure including: direct measures of physical radiation levels, expedited medical assessment (for the prodromal signs and symptoms of exposure), and measurements of surrogate effects of radiation through hematological bioindicator analysis (e.g. cytogenetic analysis of chromosomes) [1]. Individuals at risk for acute radiation syndrome, e.g. those receiving ≥ 2Gy exposure, should be rapidly identified by these approaches to determine eligibility for therapy [2].

Physical dose can be inferred from the prodromal response of absorbed radiation, which include erythema, headache, fever, lethargy, tachycardia, nausea and vomiting [1, 3]. Meanwhile, biological exposure to ionizing radiation can be assessed using dicentric chromosomes (DCs), as their frequency correlates with the radiation dose in a linear quadratic manner [4, 5]. Recent advancements in automated DC analysis have reduced the time required to accurately determine biological radiation exposures [6–10]). Nevertheless, sample preparation steps continue to introduce latency in the overall procedure, precluding comprehensive assessment of large populations. Anticipated test volumes would be significant enough to require dosimetry testing that is orders of magnitude faster than what is currently available [11]. Absolute lymphocyte depletion is a strong indicator of absorbed dose, but requires repeated sampling of a patient over several hours to days after exposure [12]. For this reason, management of acute radiation exposures has often relied upon physical dosimetry as a surrogate for biological exposures. Confounding factors may affect testing results. These include the possibility of discordant physical radiation measurements and estimated biodosimetry exposures [13, 14], and other causes of indirect predictors of exposure such as time-to-emesis (such as head trauma, or comorbid infections), thereby reducing accuracy of diagnoses [15]. More importantly, dose assessment of a large population would likely overwhelm available first responders and prevent timely triage assessment. One approach to alleviating the need to individually triage the entire population would be to survey radiation levels at a subset of locations and derive a radiation map by geostatistical analysis. This survey may involve either location-based physical dosimetry, where high-risk individuals are identified based on their proximity to fixed radiation detectors or by geospatially targeted biodosimetry. We demonstrate that combining surveys (of consistent methodology) with the geolocations of these measurements can reduce sampling requirements in population-scale radiation scenarios and would be expected to decrease total radiation exposures of first responders.

Geostatistical analysis uses regression or kriging methods to interpolate environmental measurements across a range of spatial coordinates [16]. Kriging interpolates the value of unsampled locations by computing weight linear estimates at these locations using neighboring data [17]. The mining industry application of kriging to estimate the contiguous distribution of mineral deposits from limited number of samples [18, 19] motivated us in this study to apply kriging for geographic extrapolation of absorbed radiation from a fraction of potentially exposed individuals or sampling locations. There are two variants of classical kriging, depending on whether the mean of the set of exposures is stationary (Simple Kriging) or not (Ordinary and Universal kriging) [17]. Empirical Bayesian Kriging (EBK) differs from classical kriging by using restricted maximum likelihood (REML) and accounts for measurement uncertainty, while other kriging methods use weighted least squares [20].

Our objective was to implement a geostatistical approach that accurately estimates the radiation exposures on a population-scale, based on the sampling of a subset of individuals or locations at software-guided coordinates. To validate this approach, we conducted simulated analyses of multiple population-scale nuclear detonation scenarios, using simulated dose plumes generated by HPAC (Hazard Prediction and Assessment Capability) software as ground truth estimates of exposures [21]. HPAC models the transport and dispersion of chemical, biological, radiological and nuclear releases into the atmosphere based on historical weather patterns and predicts the effects of those hazards on civilian and military populations [21]. The question this paper addresses is whether the radiation plumes derived by HPAC can be reconstructed with iterative kriging using a relatively small number of samples consistently analyzed by the same dosimetry method.

## Methods

### Overview

Nuclear detonation scenarios created by HPAC were derived for 22 North American cities and surrounding regions. Initially, locations within likely fallout areas were sampled from the potentially affected population. In each real-world scenario, a set of randomly sampled locations downwind of the event epicenter are initially selected. In each scenario, we assume the epicenter location ("ground zero"), wind direction, intensity, and weather conditions. In this paper, we refer to these locations as samples, regardless of whether the radiation is quantified from either physical emissions or by quantifying absorbed biological effects. We assume that all measurements used to derive the plume are consistently obtained by the same approach. The contours of the HPAC radiation plumes were used to specify the dose levels and random locations downwind of the epicentre. In HPAC, the contours are created from the points at locations specifying the contour threshold, in which exposure levels are known. Simulated dosimetry measurements were created from these samples and used to populate geostatistical-derived radiation maps. In the simulation, we assume that map locations between the contour boundaries exhibit the same radiation levels of the outer boundary. This introduces a source of systematic error into dose estimation, since it is likely that that the actual exposures at these locations should be interpolated between the neighboring contours that circumscribe the specific sample location. Samples were then simulated, initially at randomly selected locations, and which were then used to further refine the radiation exposure plume. After the initial set of samples were analyzed and an initial draft plume was computed by kriging, subsequent sampling locations were specified by densification. Densification is the geostatistical procedure that targets and localizes an additional small cohort of sampling locations to mitigate uncertainty in environmental measurements. These kriging and densification processes are repeated for a limited number of iterations until the coverage area and the radiation level contours of

the inferred plume stabilizes (i.e. additional sampling in the affected area does not significantly alter the geographic coverage of the plume or the estimates of absorbed radiation dose). Comparisons between independent replicates of the same scenarios using different, randomized, initial sample distributions evaluated the reliability of this approach. A detailed sequential protocol is available (http://dx.doi.org/10.17504/protocols.io.ba4nigve; "Protocol for Geostatistical Determination of Radiation Dosimetry Maps of Population-Scale Exposures").

## Acquisition and processing of United States (US) census data for geostatistical analysis

The sizes of populations impacted by simulated nuclear events in each scenario were based on 2017 US Census Data of affected counties and subdivisions. US state and sub-division boundary files (in KML [Keyhole Markup Language] format) were retrieved from the US census bureau (https://www2.census.gov/geo/tiger/GENZ2016/shp/). Population data were downloaded from the US Census ("Incorporated Places and Minor Civil Divisions Datasets: Subcounty Resident Population Estimates: April 1, 2010 to July 1, 2017"), supplemented with additional population data from the American Fact Finder and Home Town Locator (both updated July 1, 2018).

Geostatistical analysis was implemented with the ArcGIS Runtime SDK (Software Development Kit) for Java and the ArcPy package. Geostatistical mapping was performed with customized Python scripts calling the ArcGIS software toolkit (ESRI, Redlands, California, United States). For import into the ArcGIS environment, the US state sub-division boundary files were first split by sub-division name and imported to ArcMap using the 'KMLtoLayer' tool. Exceptions handled states with multiple sub-divisions that share the same name, so that ArcMap did not also select unintended sub-divisions found outside of the radiation plume. Subdivision names with spaces or dashes were also modified by concatenating them prior to 'KMLtoLayer' conversion to avoid their unintended truncation.

## Derivation of ground-truth HPAC radiation plumes

Radiological release scenarios were derived for 24 different US locations (in 22 cities) by the University of Ontario Institute of Technology (UOIT) Health Physics and Environmental Safety Research Group. Radiation plumes were simulated using HPAC v4.04 (developed by the Defense Threat Reduction Agency [DTRA]), which models the dispersion of radiation (as well as chemical and biological releases) assuming only the location of the epicenter, historical prevailing wind direction and speeds, and weather conditions. A typical ground truth plume comprised 4000 sample locations, each coincident with the contour boundaries for each radiation level. The majority of the nuclear incident scenarios did not include precipitation, except for New York and Washington D.C., where rain and snow were compared with normal plume conditions. The HPAC data are represented as a series of high-density data points at geospatial coordinates which define the shape of each radiation contour level of the plume. Topological exposure contours were computed in increments of 0.5Gy, across the 0.0–7.0Gy range, with 1.0Gy intervals shown. Contours were plotted on top of a US city map layer generated by the Humanitarian OpenStreetMap project (https://export.hotosm.org/en/v3/).

## Radiation plume reconstruction with iterative kriging and densification

The HPAC-generated plume for each scenario was visualized with the ArcMap software toolkit. The plume location was used to estimate affected population size. The US census-defined sub-divisions, which overlap and/or surround the plume of interest, were determined from overlap with their respective latitude and longitude boundaries. A Python script was

written to select data points (or 'samples') at random locations within each Census sub-division using the ArcMap tool, 'CreateRandomPoints_management'. These random samples, which corresponded to 0.1% of the population of each sub-division overlapping and surrounding the HPAC plume, simulated the locations for dose assessment. Randomization of samples was bootstrapped a minimum of 10 times per scenario to evaluate whether this procedure impacted the final plume derived by simulated physical or cytogenetic dosimetry. Results were exported using the ArcMap 'ExportXYv_stats' tool, then assigned radiation level values corresponding to the adjacent outer HPAC contour by a script comparing each sample with its location within the HPAC plume. The number of random samples generated for each scenario, and how many of those overlapped the HPAC plume, is available in S1 Table. Only a subset of the random samples shown overlapped the HPAC-derived plume; all remaining samples were considered to be unirradiated. The resulting output was then re-uploaded into ArcMap to perform kriging and densification.

We used kriging, a geostatistical interpolation technique which computes the spatial autocorrelation between data points (unlike deterministic interpolation techniques) to map predicted radiation levels by sampling around the epicentre of the event and surrounding region. Various kriging methods, available through the ArcMap extension "Geostatistical Analyst", were evaluated for this study: Ordinary, Simple, Universal, and Empirical Bayesian kriging (EBK; each kriging method described in the S1 Methods). To determine which type of kriging generated the most accurate plume, random points representing 1% of the population of the Boston (N = 617,594) and Cambridge (N = 105,162) subdivisions were generated by ArcMap, of which 223 points overlapped the Boston HPAC plume (predicted dose >0Gy). While no plume could be derived from Simple kriging, the methods of Ordinary, Universal and EBK were successful (S1 Fig). Ultimately, EBK was used for all further analyses featured in this manuscript, as the plume generated best represented the expected HPAC plume (with a 57% overlap to HPAC plume vs. 35% overlap for Ordinary and Universal) and has the advantage of taking uncertainty measurements into account (S1 Fig). The presence of unirradiated data points adjacent to the plume was found to be crucial for accurate kriging, since these points served as boundaries for kriging. A high number of unirradiated (0Gy) samples can depress the range of the plume; therefore, these locations were restricted to the subdivisions immediately surrounding the irradiated region. The number of unirradiated samples for each scenario replicate is listed in S1 Table. We envision that testing could be greatly reduced by initially measuring background or low level physical radiation in population scale events by aerial surveys or targeted multiplex dosimetry. Dose reconstruction has been extensively modeled over varied geographical, topological and weather scenarios [21]. In an actual event, environmental physical measurements at these locations without detectable radiation could substitute for dosimetry at this boundary, in order to focus attention on sampling within irradiated regions [22].

The "Densify Sampling Network" tool of the Geostatistical toolbox indicates lower confidence regions in the kriging-derived map, i.e. regions with highest variance specifying radiation dose [17]. We applied this tool to limit results to regions that would most likely exceed a pre-defined radiation level threshold. In practice, the locations selected by densification would be used to direct first responders to new locations for subsequent rounds of data acquisition in order to improve the accuracy of the kriging-derived map. Using 2Gy as the critical threshold (selection criterion QUARTILE_THRESHOLD_UPPER option), densification on one plume identified a maximum of 200 new sampling locations. We assume that locations within the 0Gy envelope surrounding the plume do not have to be sampled in subsequent kriging iterations. Densification is a compute intensive step, requiring approximately 1 hour on a desktop with an Intel i7-4770 processor [3.4Ghz] and 16GB of RAM. Note that reducing the number of

requested sampling locations decreases overall processing time. The Densify Sampling Network tool would sometimes select a sample at the same latitude and longitude between iterations. Furthermore, many densification-selected samples did not overlap the HPAC plume. As a consequence, the process often did not always yield 200 unique samples with values exceeding 0Gy. New sample data were assigned radiation values based upon their locations within the HPAC-generated plume, and kriging was performed on these and the original samples to generate another iteration of the inferred plume.

## Simulated analyses of population-scale, nuclear radiation scenarios

A geostatistical workflow managed iterative computation of the inferred radiation plume in population-scale radiation events when processing new samples (Fig 1). The purpose of simulating analyses of the radiation exposures was to determine accuracy for distinguishing clinically relevant exposure levels, and the number of irradiated individuals or locations necessary to measure dose. This was based on an initial random set of location-based data points representing sample measurements and locations collected by first responders, followed by measurements at additional locations which were assigned by densification and kriging.

EBK with default values for optional parameters was selected to predict the dose value which each location received (Fig 1), which generated a dosimetry-based radiation plume establishing the spatial boundaries of graduated exposure doses. Two consecutive plumes were

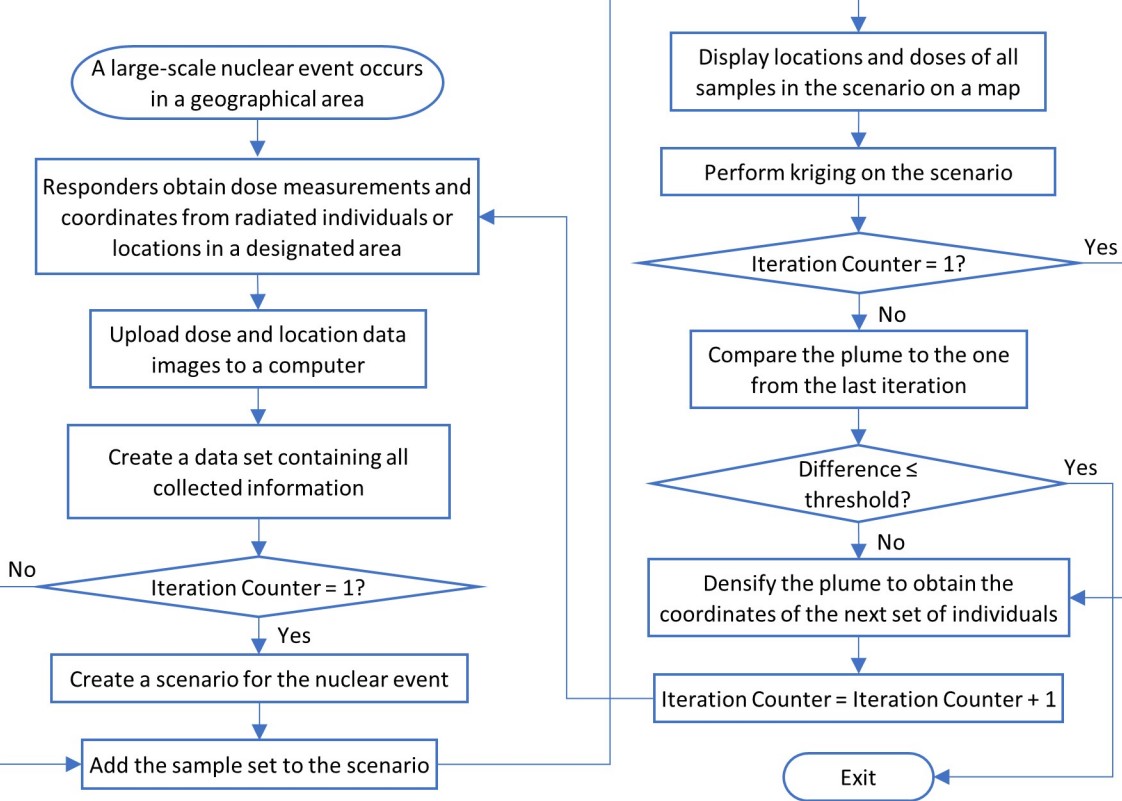

**Fig 1. The workflow for handling a population-scale radiation event.** First responders collect dose measurements and coordinates of the tested individuals or locations. Measurements are then collected and mapped. A dose plume is generated, and densification is used to select locations with lower confidence radiation estimates for follow-on sampling. These steps are repeated until output plume dosimetry levels converge. The resultant plume can be used to differentiate locations associated with significant exposures (≥2Gy) from those below this or other thresholds.

quantitatively compared with a heat map matrix with overlapping radiation levels. Each row or column respectively corresponded to a dose range in the current or previous plume, and the dose ranges were sorted in the ascending order downward or rightward. Each cell indicates the average overlap percentage between two dose ranges in terms of area. Therefore, two identical plumes resulted in an identical identity matrix. The dissimilarity between the two consecutive plumes was quantified by the diagonal Bray-Curtis dissimilarity (BCD) and root-mean-square deviation (RMSD) between the heat map matrix and the identity matrix. Lower BCD and RMSD values indicate better fit [23, 24]. For this study, BCD is computed as:

$$BCD = \frac{\sum_{i=1}^{n} |A_{ideal,i} - A_{computed,i}|}{\sum_{i=1}^{n} (A_{ideal,i} + A_{computed,i})} \tag{1}$$

where $n$ represents the 8 topological contours of the plumes (<1Gy, 1-2Gy, 2-3Gy, 3-4Gy, 4-5Gy, 5-6Gy, 6-7Gy, >7Gy), $A_{computed}$ is the percent area overlap of a particular contour of each plume, and $A_{ideal}$ represents an identical overlap between the plumes ($A_{ideal} = 1$). RMSD is computed as:

$$RMSD = \sqrt{\frac{1}{n} \sum_{i=1}^{n} (A_{ideal,i} - A_{computed,i})^2} \tag{2}$$

The diagonal values of the heat matrix represent the overlap of each radiation dose range (the topological contours) between compared plumes (see center heat matrices in Fig 2). The converging BCD and RMSD thresholds of 1/19 and 0.1 (respectively) were computed as equivalent to heat maps with a 90% overlap across each radiation dose level. This 90% stringency of overlap between consecutive iterations of geostatistical analysis was selected as a compromise to limit the number of samples while building a strong approximation of each HPAC plume (which themselves vary in shape and size). This has been a threshold indicated in prior geostatistical studies [25, 26]. The iterative workflow was discontinued when either metric dropped below the threshold.

The geostatistical workflow was used to analyze three replicates of each scenario, with each replicate initiated with a different set of random sample locations within the plume. After each iteration of kriging and densification, the resultant plumes were compared with the preceding versions (and the HPAC-based map) and presented using heat maps indicating overlap between different radiation levels. Variation between radiation levels at the same locations in successive plumes determined whether the distribution of the initially sampled locations affected the resulting maps. BCD and RMSD were computed when comparing derived plumes to both the HPAC plume (S1 Table) and all other replicates for the same scenario (S2 Table). Since the HPAC plume cannot be directly compared with the converged dosimetry plume because their data formats are incompatible, EBK was performed on all joint vertices associated with the mapped dosimetry results. This produced a topographic map that best approximated the HPAC plume. The percentage of individuals with ≤2Gy exposures that were correctly localized, assuming a uniform population distribution within each subdivision, was determined by converting the area overlap ratio with population. Area overlap is converted to estimated population affected by multiplying the area of plume overlap by the total population contained within a subdivision, divided by the area of said subdivision.

We further evaluated the proposed geostatistical approach by testing the method under 2 suboptimal sampling conditions. We mimicked improper sampling due to an inaccurate specification of wind direction for the Albany NY scenario by providing samples which partially deviated from the affected census sub-divisions (e.g. undersampling the region overlapping the plume while oversampling a non-irradiated region; S1 Table). The angle was determined

## A. New York, NY

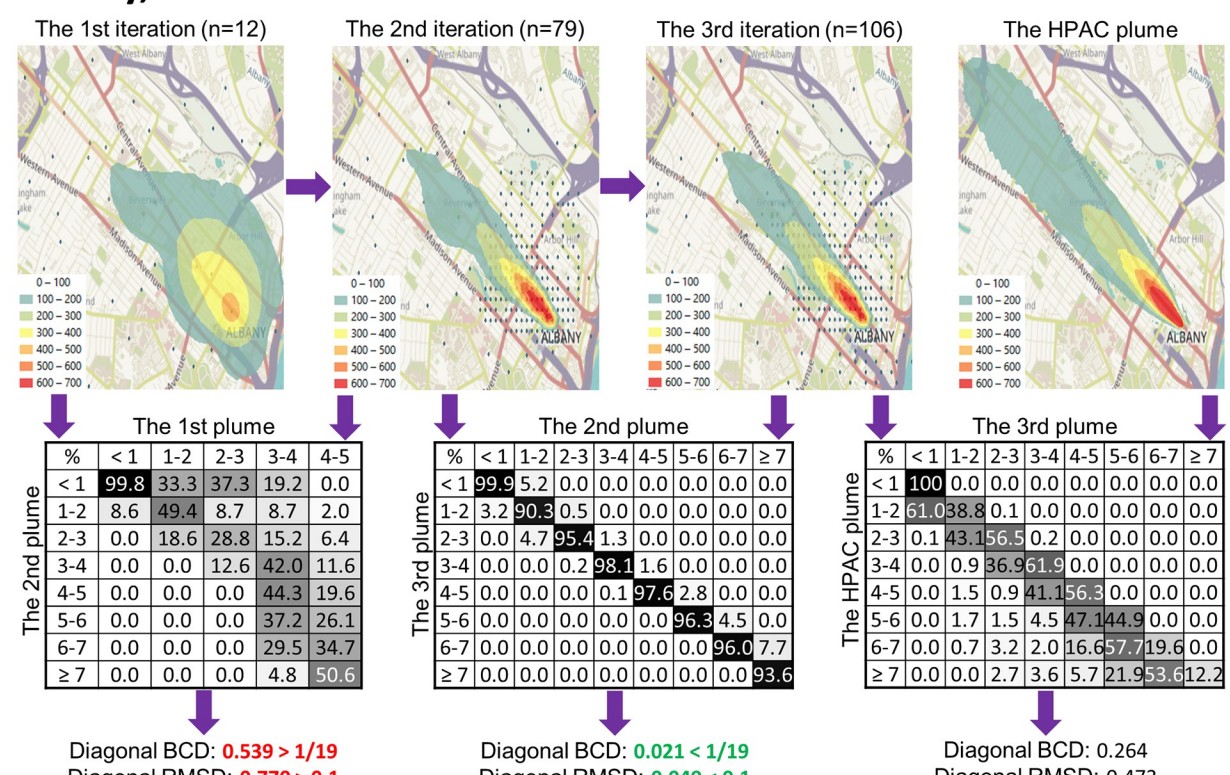

The 1st iteration (n=74)    The 2nd iteration (n=238)    The 3rd iteration (n=332)    The HPAC plume

**The 1st plume**

| The 2nd plume \ % | < 1 | 1-2 | 2-3 | 3-4 | 4-5 | 5-6 | 6-7 |
|---|---|---|---|---|---|---|---|
| < 1 | 99.9 | 7.3 | 10.0 | 3.8 | 1.3 | 0.0 | 0.0 |
| 1-2 | 0.4 | 86.7 | 11.8 | 3.0 | 1.3 | 0.0 | 0.0 |
| 2-3 | 0.0 | 8.7 | 65.5 | 12.8 | 1.5 | 0.5 | 0.0 |
| 3-4 | 0.0 | 0.0 | 4.3 | 70.1 | 3.5 | 3.6 | 0.0 |
| 4-5 | 0.0 | 0.0 | 0.5 | 32.5 | 39.2 | 5.2 | 0.0 |
| 5-6 | 0.0 | 0.0 | 0.1 | 11.1 | 60.2 | 9.4 | 0.0 |
| 6-7 | 0.0 | 0.0 | 0.0 | 5.7 | 41.7 | 29.6 | 10.6 |
| ≥ 7 | 0.0 | 0.0 | 0.0 | 0.0 | 34.9 | 28.0 | 43.7 |

Diagonal BCD: **0.354 > 1/19**
Diagonal RMSD: **0.634 > 0.1**

**The 2nd plume**

| The 3rd plume \ % | < 1 | 1-2 | 2-3 | 3-4 | 4-5 | 5-6 | 6-7 | ≥ 7 |
|---|---|---|---|---|---|---|---|---|
| < 1 | 99.9 | 1.7 | 0.0 | 0.0 | 0.0 | 0.0 | 0.0 | 0.0 |
| 1-2 | 0.1 | 96.5 | 3.7 | 0.0 | 0.0 | 0.0 | 0.0 | 0.0 |
| 2-3 | 0.0 | 3.6 | 91.8 | 3.8 | 0.0 | 0.0 | 0.0 | 0.0 |
| 3-4 | 0.0 | 0.0 | 2.6 | 94.0 | 2.1 | 0.0 | 0.0 | 0.0 |
| 4-5 | 0.0 | 0.0 | 0.0 | 2.3 | 96.0 | 0.9 | 0.0 | 0.0 |
| 5-6 | 0.0 | 0.0 | 0.0 | 0.0 | 1.6 | 97.6 | 0.2 | 0.0 |
| 6-7 | 0.0 | 0.0 | 0.0 | 0.0 | 0.0 | 3.3 | 96.6 | 0.0 |
| ≥ 7 | 0.0 | 0.0 | 0.0 | 0.0 | 0.0 | 0.0 | 2.5 | 98.2 |

Diagonal BCD: **0.018 < 1/19**
Diagonal RMSD: **0.043 < 0.1**

**The 3rd plume**

| The HPAC plume \ % | < 1 | 1-2 | 2-3 | 3-4 | 4-5 | 5-6 | 6-7 | ≥ 7 |
|---|---|---|---|---|---|---|---|---|
| < 1 | 100 | 0.0 | 0.0 | 0.0 | 0.0 | 0.0 | 0.0 | 0.0 |
| 1-2 | 59.2 | 40.7 | 0.0 | 0.0 | 0.0 | 0.0 | 0.0 | 0.0 |
| 2-3 | 0.0 | 46.2 | 53.7 | 0.0 | 0.0 | 0.0 | 0.0 | 0.0 |
| 3-4 | 0.0 | 0.0 | 49.2 | 50.7 | 0.0 | 0.0 | 0.0 | 0.0 |
| 4-5 | 0.0 | 0.0 | 0.6 | 48.5 | 50.8 | 0.0 | 0.0 | 0.0 |
| 5-6 | 0.0 | 0.0 | 0.0 | 1.6 | 49.8 | 48.4 | 0.0 | 0.0 |
| 6-7 | 0.0 | 0.0 | 0.0 | 1.3 | 3.0 | 82.4 | 13.2 | 0.0 |
| ≥ 7 | 0.0 | 0.0 | 0.0 | 0.0 | 0.6 | 3.9 | 12.5 | 52.4 | 30.4 |

Diagonal BCD: 0.347
Diagonal RMSD: 0.565

## B. Albany, NY

The 1st iteration (n=12)    The 2nd iteration (n=79)    The 3rd iteration (n=106)    The HPAC plume

**The 1st plume**

| The 2nd plume \ % | < 1 | 1-2 | 2-3 | 3-4 | 4-5 |
|---|---|---|---|---|---|
| < 1 | 99.8 | 33.3 | 37.3 | 19.2 | 0.0 |
| 1-2 | 8.6 | 49.4 | 8.7 | 8.7 | 2.0 |
| 2-3 | 0.0 | 18.6 | 28.8 | 15.2 | 6.4 |
| 3-4 | 0.0 | 0.0 | 12.6 | 42.0 | 11.6 |
| 4-5 | 0.0 | 0.0 | 0.0 | 44.3 | 19.6 |
| 5-6 | 0.0 | 0.0 | 0.0 | 37.2 | 26.1 |
| 6-7 | 0.0 | 0.0 | 0.0 | 29.5 | 34.7 |
| ≥ 7 | 0.0 | 0.0 | 0.0 | 4.8 | 50.6 |

Diagonal BCD: **0.539 > 1/19**
Diagonal RMSD: **0.770 > 0.1**

**The 2nd plume**

| The 3rd plume \ % | < 1 | 1-2 | 2-3 | 3-4 | 4-5 | 5-6 | 6-7 | ≥ 7 |
|---|---|---|---|---|---|---|---|---|
| < 1 | 99.9 | 5.2 | 0.0 | 0.0 | 0.0 | 0.0 | 0.0 | 0.0 |
| 1-2 | 3.2 | 90.3 | 0.5 | 0.0 | 0.0 | 0.0 | 0.0 | 0.0 |
| 2-3 | 0.0 | 4.7 | 95.4 | 1.3 | 0.0 | 0.0 | 0.0 | 0.0 |
| 3-4 | 0.0 | 0.0 | 0.2 | 98.1 | 1.6 | 0.0 | 0.0 | 0.0 |
| 4-5 | 0.0 | 0.0 | 0.0 | 0.1 | 97.6 | 2.8 | 0.0 | 0.0 |
| 5-6 | 0.0 | 0.0 | 0.0 | 0.0 | 0.0 | 96.3 | 4.5 | 0.0 |
| 6-7 | 0.0 | 0.0 | 0.0 | 0.0 | 0.0 | 0.0 | 96.0 | 7.7 |
| ≥ 7 | 0.0 | 0.0 | 0.0 | 0.0 | 0.0 | 0.0 | 0.0 | 93.6 |

Diagonal BCD: **0.021 < 1/19**
Diagonal RMSD: **0.049 < 0.1**

**The 3rd plume**

| The HPAC plume \ % | < 1 | 1-2 | 2-3 | 3-4 | 4-5 | 5-6 | 6-7 | ≥ 7 |
|---|---|---|---|---|---|---|---|---|
| < 1 | 100 | 0.0 | 0.0 | 0.0 | 0.0 | 0.0 | 0.0 | 0.0 |
| 1-2 | 61.0 | 38.8 | 0.1 | 0.0 | 0.0 | 0.0 | 0.0 | 0.0 |
| 2-3 | 0.1 | 43.1 | 56.5 | 0.2 | 0.0 | 0.0 | 0.0 | 0.0 |
| 3-4 | 0.0 | 0.9 | 36.9 | 61.9 | 0.0 | 0.0 | 0.0 | 0.0 |
| 4-5 | 0.0 | 1.5 | 0.9 | 41.1 | 56.3 | 0.0 | 0.0 | 0.0 |
| 5-6 | 0.0 | 1.7 | 1.5 | 4.5 | 47.1 | 44.9 | 0.0 | 0.0 |
| 6-7 | 0.0 | 0.7 | 3.2 | 2.0 | 16.6 | 57.7 | 19.6 | 0.0 |
| ≥ 7 | 0.0 | 0.0 | 2.7 | 3.6 | 5.7 | 21.9 | 53.6 | 12.2 |

Diagonal BCD: 0.264
Diagonal RMSD: 0.473

**Fig 2.** The simulated analysis of the no-precipitation scenarios of urban New York [Replicate #4] (A) and Albany [Replicate #1] (B). Each of the three left-most maps (first row) display sample locations and EBK-derived dosimetry plumes after several rounds of densification (iterations; note gradual improvement of plume from left to right), while the right-most map displays the HPAC plume. Plume colours (from red to purple) indicate the dose ranges of the region (from <1Gy [100cGy] to ≥7Gy [700 cGy]). The total number of samples exceeding 0Gy in each iteration are indicated (in parentheses). The matrices (second row) are used to compare two plumes at each dose range, where each value indicates the area overlap (background grey level is proportional to the percent overlap) of two regions of plume pair. The right-most matrix compares the final derived plume to the HPAC plume. Below the matrices, the diagonal BCD and RMSD values between successive iterations represented by each heat matrix are indicated. The green or red colors indicate that these two metrics exceeded or did not meet the threshold for convergence of the procedure, respectively. Subsequent iterations gradually improved the computed radiation dosimetry plume, with each adding a small number of new samples. Based on the census data, the converged plumes localized 80.3% and 75% of the locations (weighted for population density) for treatment-eligible radiation exposures in these scenarios.

by finding the distance of mean latitude or longitude of all random points and computing an angle against the direction of the HPAC plume. To simulate variation in the precision of the radiation dose measurements, we applied random dose errors to HPAC generated radiation doses (to a maximum value of either ± 0.5Gy or ± 1.0Gy) to all initial and densification-selected samples. A representative subset of the previously analyzed radiation release scenarios was assessed for both plume accuracy and for the numbers of kriging/densification iterations and samples required for convergence by iterative kriging and densification (S3 Table). These included Birmingham AL (Replicate #2), Boston MA (Replicate #1), Chicago IL (Replicate #1), Columbia SC (Replicate #1), and Columbus OH (Replicate #1 and 2).

## Results

### Development of simulated radiation plumes

We simulated the analysis of 28 population-scale, 10 megaton yield nuclear detonation scenarios with dosimetry data for radiation exposures corresponding to HPAC-derived dose estimates. These results were used to derive plumes of absorbed radiation exposures. The simulations included 22 in urban/high-density populated regions of the United States, and 2 rural/low-density scenarios. The process of kriging and densification was repeated until the difference between the current and previous plume reached a minimum threshold (Fig 2). The number of samples with radiation exposure used to generate the final iteration of a plume would vary between different scenarios (ranging from 58 and 347 samples within the irradiated plume; starting from an initial sample set representing 0.1% of the population; Table 1). For less densely populated regions, initial sampling was also performed at higher population densities (0.2 and 1.0%). The number of samples necessary to reach this stopping point was sometimes variable between replicates, and this variability was inversely proportional to the overall population density of the region (from the US census). Cities with a high population density (>10,000 per square mile) had a 16.0% average coefficient of variation (CV) of the number of samples necessary to reach convergence, whereas the scenarios within low-density regions (<10,000 per square mile) had an average CV of 21.4% (see S4 Table for all CVs). In general, more samples would lead to a plume which better resembles the HPAC result (Table 1). There are exceptions where equivalent accuracy is obtained with fewer overall points (e.g. Buffalo NY; Table 1) which implies that the spatial distribution of initial set of random samples can also influence the accuracy of the converged plume.

Success in reconstructing simulated plumes from dosimetry data was based on the accuracy of predicting irradiated samples ≥ 2Gy (as defined by the HPAC plume). This radiation level threshold was selected based on US government recommendations for eligibility of clinical treatment of Acute Radiation Syndrome by cytokine therapies [27]. Results are reported based on the accuracy in distinguishing samples exposed to this threshold or higher from false positives or negatives. In these simulations, we have determined that false positive samples (i.e. ≤

**Table 1. Simulated analyses of population-scale nuclear radiation scenarios on 22 cities.**

| Scenario | | | Replicate: | No. of Iterations[2] | No. of Samples (>0Gy) | Accuracy (%) | |
|---|---|---|---|---|---|---|---|
| City | Population Density[1] | Special Conditions | | | | ≥ 2Gy | ≥ 3Gy |
| Albany, NY | Urban | None | 1 | 3 | 106 | 75.0 | 83.2 |
| | | | 2 | 4 | 119 | 78.4 | 84.5 |
| | | | 3 | 5 | 186 | 76.1 | 82.7 |
| Alexandria, VA | Urban | None | 1 | 4 | 137 | 57.8 | 83.3 |
| | | | 2 | 3 | 129 | 56.0 | 81.2 |
| | | | 3 | 3 | 127 | 52.8 | 79.3 |
| Baltimore, MD | Urban | None | 1 | 4 | 111 | 75.0 | 80.2 |
| | | | 2 | 4 | 115 | 75.9 | 83.9 |
| | | | 3 | 3 | 111 | 76.2 | 77.0 |
| Birmingham, AL | Urban | None | 1 | 4 | 98 | 67.8 | 43.6 |
| | | | 2 | 4 | 101 | 69.6 | 39.7 |
| | | | 3 | 4 | 87 | 71.9 | 49.3 |
| Boston, MA | Urban | None | 1 | 5 | 166 | 67.7 | 84.7 |
| | | | 2 | 3 | 132 | 59.2 | 85.3 |
| | | | 3 | 3 | 143 | 63.4 | 83.3 |
| Buffalo, NY | Urban | None | 1 | 6 | 111 | 66.8 | 84.3 |
| | | | 2 | 6 | 222 | 64.6 | 77.7 |
| | | | 3 | 4 | 138 | 62.5 | 82.0 |
| Burlington, VT | Urban | None | 1 | 4 | 177 | 90.2 | 88.6 |
| | | | 2 | 5 | 205 | 90.3 | 85.1 |
| | | | 3 | 7 | 205 | 90.7 | 88.1 |
| Camden, NJ | Urban | None | 1 | 5 | 136 | 71.1 | 54.9 |
| | | | 2 | 5 | 142 | 70.8 | 62.2 |
| | | | 3 | 8 | 180 | 71.6 | 62.6 |
| Charleston, SC | Urban | None | 1 | 4 | 66 | 60.5 | 67.8 |
| | | | 2 | 4 | 66 | 58.5 | 73.7 |
| | | | 3 | 4 | 94 | 70.7 | 65.8 |
| Charlotte, NC | Urban | None | 1 | 5 | 133 | 67.5 | 72.2 |
| | | | 2 | 4 | 77 | 60.9 | 76.9 |
| | | | 3 | 7 | 114 | 65.6 | 76.5 |
| Chicago, IL | Urban | None | 1 | 4 | 63 | 78.7 | 54.8 |
| | | | 2 | 4 | 59 | 70.7 | 37.9 |
| | | | 3 | 4 | 62 | 78.2 | 54.4 |
| Cincinnati, OH | Urban | None | 1 | 6 | 139 | 73.2 | 72.0 |
| | | | 2 | 8 | 144 | 72.5 | 64.8 |
| | | | 3 | 9 | 288 | 69.6 | 70.3 |
| Cleveland, OH | Urban | None | 1 | 5 | 130 | 72.0 | 77.2 |
| | | | 2 | 4 | 128 | 73.2 | 78.8 |
| | | | 3 | 4 | 143 | 72.2 | 77.7 |
| Columbia, SC | Urban | None | 1 | 5 | 126 | 66.0 | 26.7 |
| | | | 2 | 4 | 72 | 30.1 | 3.9 |
| | | | 3 | 4 | 75 | 60.5 | 30.4 |
| | | | 4 | 4 | 78 | 55.6 | 2.6 |
| | | | 5 | 3 | 79 | 27.6 | 0.0 |
| | | 0.2% population[3] | 6 | 6 | 218 | 65.1 | 48.2 |
| | | | 7 | 3 | 114 | 58.1 | 52.9 |
| | | 1% population[3,4] | 8 | 7 | 1409 | 69.8 | 72.7 |

*(Continued)*

**Table 1.** (Continued)

| Scenario | | | Replicate: | No. of Iterations[2] | No. of Samples (>0Gy) | Accuracy (%) | |
|---|---|---|---|---|---|---|---|
| City | Population Density[1] | Special Conditions | | | | ≥ 2Gy | ≥ 3Gy |
| Columbus, OH | Urban | None | 1 | 4 | 74 | 66.5 | 74.1 |
| | | | 2 | 4 | 137 | 44.1 | 19.0 |
| | | | 3 | 6 | 123 | 74.7 | 73.8 |
| | | | 4 | 8 | 130 | 67.9 | 71.1 |
| | | | 5 | 5 | 103 | 66.5 | 72.2 |
| | | 0.2% population[3] | 6 | 4 | 378 | 68.8 | 78.6 |
| Des Moines, IA | Urban | None | 1 | 4 | 240 | 74.7 | 63.5 |
| | | | 2 | 5 | 282 | 70.2 | 60.8 |
| | | | 3 | 4 | 271 | 77.0 | 62.7 |
| Detroit, MI | Urban | None | 1 | 4 | 117 | 80.3 | 74.3 |
| | | | 2 | 5 | 132 | 76.2 | 74.6 |
| | | | 3 | 4 | 123 | 79.2 | 77.7 |
| Evansville, IN | Urban | None | 1 | 3 | 76 | 67.8 | 45.1 |
| | | | 2 | 5 | 68 | 63.4 | 39.1 |
| | | | 3 | 5 | 58 | 70.6 | 47.0 |
| Grand Rapids, MI | Urban | None | 1 | 9 | 199 | 72.9 | 73.8 |
| | | | 2 | 4 | 168 | 74.5 | 74.7 |
| | | | 3 | 6 | 156 | 73.7 | 75.5 |
| New York, NY | Rural | None | 1 | 8 | 131 | 68.8 | 63.7 |
| | | | 2 | 10 | 181 | 69.0 | 66.8 |
| | | | 3 | 5 | 101 | 64.9 | 72.1 |
| | Urban | None | 1 | 3 | 308 | 80.3 | 83.4 |
| | | | 2 | 3 | 347 | 81.4 | 82.1 |
| | | | 3 | 3 | 332 | 81.0 | 80.5 |
| | | | 4 | 3 | 332 | 80.8 | 80.5 |
| | | Rain | 1 | 3 | 265 | 72.3 | 86.4 |
| | | Snow | 1 | 3 | 275 | 74.3 | 87.4 |
| Philadelphia, PA | Urban | None | 1 | 4 | 79 | 79.2 | 75.0 |
| | | | 2 | 7 | 91 | 75.5 | 63.5 |
| | | | 3 | 5 | 79 | 74.9 | 68.5 |
| Washington D.C. | Rural | None | 1 | 3 | 120 | 74.7 | 74.8 |
| | | | 2 | 4 | 148 | 73.6 | 73.7 |
| | | | 3 | 7 | 142 | 73.1 | 71.5 |
| | Urban | None | 1 | 5 | 237 | 68.2 | 82.6 |
| | | | 2 | 3 | 190 | 66.9 | 82.2 |
| | | | 3 | 3 | 193 | 66.9 | 79.5 |
| | | Rain | 1 | 5 | 237 | 68.2 | 82.6 |
| | | Snow | 1 | 3 | 131 | 82.0 | 83.7 |

[1] 'Urban' and 'Rural' indicate whether the HPAC plume was placed in a region of high or low population density, respectively. For example, while both Washington DC scenarios have the same point of origin within Washington sub-division, the low-density scenario encompasses a region southwest of the city including Arlington National Cemetery.

[2] The number of iterations (kriging and densification steps) required to reach stopping criteria for this replicate.

[3] Plumes derived with increased initial sampling rate (representing 0.2% or 1.0% of the population of each subdivision) under normal weather conditions.

[4] The densification threshold for the 1.0% Columbia SC replicate was increased to allow for >200 samples.

2Gy account for [on average] 0.5─2% of those estimated to be exposed at ≥2Gy. On average, the accuracy of the simulated plumes to predict exposures above this threshold (relative to the HPAC radiation plume) was 69.8%, ranging from 52.8% (Alexandria VA) to 90.7% (Burlington VT). The accuracy range for replicates from the same geographic region was also consistent, i.e. within 10% difference between replicates.

Two of the 28 scenarios exhibited outlier replicates, in which accuracy was impacted by the random sampling procedure and local population densities (Columbia SC replicate #2 [30.1%] and Columbus OH replicate #2 [44.1%]; Table 1 and S1 Table). Additional replicates were analyzed for these scenarios. While both additional replicate sets for the Columbus OH scenario closely resembled the first replicate (67–68% accuracy), one of the Columbia SC replicates continued to perform poorly (27.6% [replicate #5]). Further investigation suggested that this result might not correlate with the number of initial sampled locations within the irradiated region (12–16 samples exceeding 0Gy). Rather, it was apparent that the initial sets of sample data for these particular replicates exhibited sparse coverage over large regions of the HPAC radiation plume. Including subsequent densification steps did not always identify additional sample locations that resulted in a contiguous 2Gy plume similar to the entire HPAC plume. This issue was addressed by increasing the population fraction that was initially sampled. By combining the initial samples of the two underperforming Columbia SC replicates (#2 and #5; now representing 0.2% of the population rather than 0.1%), a plume was generated with accuracy equivalent to the best performing Columbia replicate (65.1% accuracy in 6 iterations [replicate #6]; Table 1). An independently selected random set of samples representing 0.2% of the population led to a comparable (albeit slightly lower) accuracy rate using fewer kriging and densification cycles (58.1% in 3 iterations [replicate #7]). A further increase in sampling rate to 1.0% of the population improved performance relative to the best Columbia SC replicate (+3.8% more accurate; replicate #8). Thus, in scenarios with low population densities, the population fraction sampled will need to be increased to construct a contiguous dosimetry map that resembles the actual radiation plume.

In rare cases, low population scenarios would fail to yield a plume in the initial step, regardless of kriging method used. When this occurred, the iteration methods could not progress to the following densification steps, and plume development was halted. The numbers of samples which overlapped the HPAC plume were found to be extremely low in these cases, ranging from 0–2 samples in total. There are instances in which 2 irradiated samples were adequate to progress plume development (e.g. Cincinnati urban sampling #2 [Table 1]). In these cases, a greater number of iterations were required for plume development to fulfill the established stopping criteria (N = 8 iterations for the Cincinnati example). Therefore, while the minimum number of irradiated samples can be extremely low, development of accurate dosimetry plumes in such cases could require unacceptable levels of exposure for first responders.

We also compared coverage of derived plumes at a higher radiation level (≥ 3Gy), since these were expected to be more compact than the ≥ 2Gy contour, more densely populated, and more likely to result in severe clinical symptoms. Based on the degree of overlap between the derived and HPAC plumes (Table 1), 19 of 28 scenarios were more accurate at the ≥ 3Gy contour than at ≥ 2Gy level for at least one replicate (greater accuracy was seen for all replicates of 13 scenarios). In particular, the Alexandria VA, Boston MA and Buffalo NY cases respectively exhibited an average of +25.7%, +21.0% and +16.7% higher accuracies at the 3Gy threshold. In several instances, the performance of replicates of the same scenario varied (e.g. Charleston SC replicates #2 (+15.2%) and #3 (-4.9%) in comparison with the 2Gy threshold). Derivation of plumes for the Charleston scenario has been challenging due to differences in population density for different initial sampling locations, for example, replicate #5. However,

this issue was mitigated by sampling at higher densities (replicates #6–8), for example, the 1.0% population density replicate, in which the 3Gy contour accuracy exceeded that of the 2Gy contour (+2.9%).

## Distinct initial sampling locations produced similar plumes for the same scenarios

In each case, determination of the distribution of radiation exposure in each scenario began with an initial set of sampling locations within the region of the simulated radiation release scenario. The majority of these locations did not overlap with the HPAC plume and have therefore been modelled as unirradiated samples. As a result, the initial number of irradiated samples can vary significantly among different scenarios, based on the population of the region and overall plume size. Thus, the number of iterations of densification required to maximize the accuracy of the simulated dose plumes varied significantly between replicates (between 3 and 10 iterations per replicate; Table 1).

The initial sets of samples were randomly assigned locations within each census sub-division by ArcMap. We first determined whether this process affected accurate derivation of the plume as a consequence of the variable locations of these initial conditions. Replicate dissimilarity was quantified as BCD and RMSD (S2 Table) to determine the extent to which this source of variation could impact the accuracy of the derived radiation plumes. The BCD between plume replicates were less than 0.3 (or a >70% similarity between them [23]) for nearly all scenarios (S2 Table). New York NY and Cincinnati OH show > 90% similarity (by BCD) across all replicates, while Albany NY, Boston MA, Birmingham AL, Camden NJ, Cleveland OH, Detroit MI, Grand Rapids MI, Philadelphia PA and Washington D.C. (urban scenario) have at least two replicates with a >90% similarity to each other.

## Increasing stringency of overlap between consecutive iterations of radiation plumes

To expedite dose estimation for triage management of a nuclear incident, sampling was discontinued when the estimated populations in areas covered consecutive plume iterations were within 90% of each other based on similarity. We investigated whether further cycles of geostatistical processing would stabilize or improve the derived plumes relative to the HPAC standard. We therefore mandated a more stringent threshold for consistent coverage of iterative plumes, i.e. from 90% to 99%, and determined whether the additional iterations would more closely resemble the gold standard for one replicate of each scenario (S5 Table). On average, an additional 3–4 iterations were necessary to reach this threshold, with an average of 65 additional sampling locations. In 3 scenarios, the plumes improved significantly: rural New York (+9.6%); Alexandria VA (+7.7%); and Columbus OH (+5.1%). In the rural New York scenario, the 1Gy and 2Gy contours are significantly expanded with the inclusion of additional data points (S2 Fig). However, altering the stopping criteria did not significantly improve plume accuracy in most replicates tested (+1.2% improvement, on average). Frequently, the plume would prematurely stabilize once densification failed to reveal new sampling locations. The addition of further iterations did not always improve accuracy at the 2Gy threshold. For example, addition of 43 new unique sampling locations slightly reduced the accuracy (-2.5%) of one Boston MA replicate (S5 Table). While the 2Gy contour in this instance did expand geographically, new 1.0 and 1.5Gy samples introduced by densification made the 2Gy contour discontinuous, explaining the reduction in accuracy (S3 Fig). Contours at higher radiation levels for this plume were unaffected, and were contiguous, however.

## Impact of weather effects on derived plumes

The effects of adverse and normal weather conditions (i.e. no precipitation, rain and snow) were compared for both urban and rural New York and Washington DC scenarios (Table 1 and S1 Table). The HPAC software was used to simulate the effects of rain and snow conditions in these scenarios at the same nuclear yields. The resulting HPAC plumes closely resembled the morphology, extent, and wind vector of the no precipitation scenario. At the >2Gy threshold, both final New York plumes with precipitation exhibited less accurate geographic coverage (72% rain and 74% snow) than under normal conditions (76–81% without weather effects; S2 Table). By contrast, the Washington D.C. urban snow scenario significantly outperformed both no precipitation and rain scenarios (82% accuracy versus 66–68% for all other weather conditions; S2 Table). These plumes were found to have a 70–76% similarity to the HPAC plume (by BCD), similar to that of the same scenarios in the absence of weather effects (S1 Table). In these nuclear event scenarios, adverse weather events did not confound or impact the accuracy of population scale geographic dosimetry using the same approach described for normal weather conditions.

## Inferred radiation exposures under suboptimal sampling conditions

If the information about wind direction and location of the epicenter of a nuclear event is limited and/or inaccurate, it might be expected to lead to improper sampling. Initial samples for the Albany NY scenario were used as input, undersampling the Albany sub-division (which overlaps much of the plume) while oversampling its neighboring sub-division, Colonie, which is due north of the epicentre of the simulated nuclear event. The bearing angle of the Albany NY scenario is N 51.9º W. For the initial sample data, the populations representing Albany and Colonie were shifted from equal 0.1% proportions of their respective populations to unequal distributions of either 0.05:0.2% or 0.01:1.0%. The 0.05:0.2% ratio simulates a wind measurement error of 29.1º north (or N 22.8º W) relative to the actual wind direction that produced the HPAC plume. The 0.01:1.0% ratio corresponds to a deviation of 40.9º north (or N 11.0º W). Despite this initial sampling error, inferred radiation plumes comparable to the correct plume were obtained. The accuracies of the >2Gy threshold maps were, respectively, 74.2% for the 0.05:0.2% population ratio and 74.5% for the 0.01:1.0% ratio, which compares well with the correctly sampled initial map of 75.0–78.4%. Reconstruction of the geostatistical dosimetry map required 4 iterations with sampling error, which was only one more cycle of densification-sample procurement and kriging than the map generated from the original (correctly sampled) Albany scenario. This demonstrates the robustness of the densification step to select relevant sample points at subsequent iterations of kriging and densification that result in a correct dosimetry map. It appears that, in some scenarios, reconstruction of the original HPAC map can largely succeed and achieve convergence with fewer irradiated samples, at the expense of a single additional sampling cycle (e.g. iteration).

The original set of simulations presented do not account for error in dose estimates, which can vary considerably based on the methodology used [28]. The impact of random variation on measured dose was examined by deriving plumes with modified sample exposures with added or subtracted random errors. Random errors in radiation exposure (either ±0.5Gy or ±1.0Gy) were applied to both the initial samples and those obtained from subsequent densification steps. Maximum deviations were designed to represent confidence values in physical and/or biodosimetry methods, including physical dose estimation error [29], dicentric chromosome analysis (DCA; ±0.5Gy) or cytokinesis-blocked micronucleus assays (±1.0Gy), which exhibit higher variance in estimating dose compared to DCA [28, 30, 31]. The introduction of such measurement errors led to plumes that were generally less accurate, required a greater

number of samples, and additional iterations of kriging and densification to achieve convergence (S3 Table). The degree of error in this analysis was also correlated with decreased accuracy and increased processing time (±0.5Gy error-derived plumes were more accurate than ±1.0Gy error-derived plumes in nearly all scenarios; S3 Table). We also note that dose modifications (±1.0Gy) for the Columbia SC scenario #1 completely failed to achieve complete plume coverage, and as a consequence, did not successfully derive a plume with accurate radiation exposure levels >2Gy.

## Discussion

A large-scale nuclear detonation or radiation accident would be expected to place excessive demands on first responders to rapidly identify those individuals with significant exposures. We describe a geostatistical method of localizing significant exposures based on a significantly reduced number of tested samples or dose measurements. Overall, plumes from 28 distinct scenarios simulating absorbed radiation identified 70±10% of locations where the population are expected to receive ≥2Gy of radiation exposure (and thus are eligible for cytokine therapy), each based on between 58 and 347 samples (median of 131; excluding scenarios sampled initially at >0.1% of the population). Although initial sampling represented 0.1% of the population of the region, very few of these samples would have been exposed to radiation. The relatively high fidelity of the radiation dosimetry maps is attributable to the reconstruction of the radiation plume using geostatistical analysis of limited acquisition of additional samples at key locations within the areas of high radiation exposure. After a nuclear incident, processing all individuals for dose assessment has been acknowledged to be labor intensive, and would likely be a major bottleneck in identifying those who require immediate treatment [32].

The distribution of initial sampling seemed to have a significant impact on the overall accuracy of the modeled plume, especially in regions of low population density. In the Columbus OH and Columbia SC scenarios, one of three replicates exhibited significantly lower resemblance to the HPAC plume than the others. Performing additional replicates resulted in a second poorly performing Columbia replicate. In each replicate, a similar number of initial samples were found to be within the HPAC plume and did not correspond to the observed performance differences. Visually, the poor performance of these replicates was apparently due to a lack of coverage of large segments of the HPAC plume due to undersampling of regions of low population density, thereby making the subsequent densification steps ineffective. This was addressed by increasing the fraction of the population that was initially sampled. Thus, implementation of an initial sampling strategy that takes population density into account (e.g. maintaining an even distribution of samples) would increase the likelihood of deriving an accurate plume with fewer iterations. Additional densification steps with different settings (i.e. setting a lower radiation level threshold) could also lead to a more representative sample distribution and final distribution, however this also increases processing time (~1 hour per iteration). In a real-world scenario, secondary sampling locations assigned by densification would be supervised, which would direct the software towards derivation of a complete and accurate plume. Indeed, we found that manually adding two new sample locations to the unrepresented region corrected the poor results obtained for the Columbia SC scenario replicate after two additional cycles of kriging and densification.

The HPAC source and geostatistically-derived plumes are based on very distinct sample distributions. HPAC data are highly deterministic, with samples being densely arrayed at each respective contour boundary, whereas iterative kriging and densification distributes these sample locations throughout these regions. Kriging computes and derived plume contours from distributed sample data, but the sample locations themselves rarely coincide with HPAC

contours. At lower radiation exposures, kriging tends to produce contours that do not fully overlap the corresponding HPAC-defined plume boundaries. These regions of the derived plume do not exceed the 2Gy threshold (e.g. S3 Fig, middle plume). In this case, a discontinuity at the $\geq 2$ Gy contour is eliminated at $\geq 1.9$Gy, where 83% of the plume area overlapped or was within 0.1Gy of the $\geq 2$Gy threshold. This indicates that the derived plumes can generally approximate the HPAC plume, regardless of the accuracy at the $\geq 2$ Gy threshold. Additional improvements may rely on manual selection of sampling locations at or close to locations circumscribing the $\geq 2$ Gy threshold.

One limitation of this study is the static nature of the derived dosimetry plume presented here, that is, it is based on sampling at a single time point immediately following the nuclear incident. Radiation dispersion and decay are known to have significant effects for hours after an event [33]. Physical dosimeters deposited by radio-controlled drones, for example, at locations specified by geostatistics could provide continuous radiation levels for accurate dynamic modelling of plume evolution. Alternatively, these data could be captured and transmitted by first responders through the RadResponder Network ([34] https://www.radresponder.net/). Furthermore, since our sampling methods are based on population census data over geographic county subdivisions, they do not have sufficient granularity to model how population densities vary over the time continuum (for example, the population of Manhattan, New York City doubles during daytime hours [2013 Census estimate]). This study does not correct for the cumulative exposure, which would be particularly relevant to the individuals sheltering in radiation-contaminated areas, who may have been sampled for the creation of dosimetry maps. Finally, neither the HPAC version available for this study, nor our geostatistical models account for shielding by infrastructure, such as shadowing, which computes the degree to which radiation is prevented from reaching certain locations by the urban environment [35]. While these factors will impact the predicted accuracies of derived dosimetry maps, these effects will also have to be accounted for in ground truth models, such as HPAC, before they could be addressed in geostatistical interpolation.

This approach made several reasonable assumptions that are necessary to perform the simulation, but these also affected the accuracy of the derived radiation plumes. Currently, the radiation levels determined by dosimetry are expected to approximate full exposures and do not account for partial body exposures. These simulations were intentionally designed to avoid a circular argument that geostatistical estimation of radiation plumes were based solely on the original dose estimates; thus, the values used did not specify the same radiation dose as an interpolated value at that location (which would be expected to reconstruct the same dose). Consequently, the contours of the derived plume, while closely resembling the original HPAC thresholds, are nevertheless different. This may be the likely explanation for why the derived plume does not encompass all individuals expected to be exposed to $\geq 2$Gy of ionizing radiation in these scenarios. We also assumed that estimated doses would be sufficiently accurate to derive the radiation maps derived from those estimates.

Biodosimetry estimates absorbed radiation exposures, whereas physical dosimetry measures environmental emissions. Physical dosimetry is more rapid and can map changing radiation plume locations dynamically. However, unfiltered radiation emissions are prone to false positive readouts, for example in aerial physical dosimetry counterterrorism surveys [29] due to common environmental sources of radiation. Uncorrected, such data will introduce errors and distort geostatistically derived plumes. Mitigation may be possible by specifying the locations of radiation detectors by iterative kriging and densification. Nevertheless, biodosimetry at specified locations may provide results that might be useful for assessing treatment-eligibility in instances of borderline clinical exposures.

Simulations with increased dose estimation errors showed that the amounts of time, numbers of samples, and field excursions to procure samples would be increased in order to infer approximate exposure levels by geographic interpolation. These error sources were also responsible for the failure to derive contiguous dosimetry maps, especially in less densely populated regions. Sampling estimates are also based on the number of individuals or locations with detectable exposures. Samples outside of the plume region where radiation was not detectable were assumed to be unexposed. Nevertheless, in an actual event, some of the samples obtained might be unirradiated, which could therefore increase the number of tested samples necessary for derivation of an accurate plume.

The approach described here suggests the feasibility of quantifying radiation exposures at untested locations using either bio- or physical dosimetry. The benefits of geostatistical biodosimetry would be minimal, however, if it were possible to acquire, process and analyze large numbers of samples quickly. Processing and analysis of samples from all known or suspected irradiated individuals for biodosimetry is too labor intensive, a significant bottleneck in identifying treatment-eligible exposures [32]. While rapid interpretation of cytogenetic biodosimetry data is feasible [36], sample acquisition and data generation exceed the capacities of small teams of first responders and individual testing laboratories [37]. The proposed approach may partially overcome capacity resource limitations of first responders and biodosimetry laboratories to provide data for triage assessment of entire populations. Laboratory contexts, where sample preparation, imaging and DCA can be highly automated and multiplexed may have sufficient throughput [10, 35, 38]. Field sampling, laboratory and computational resources could be amplified through simultaneous deployment of multiple dedicated teams. With high performance computing and parallel processing [36], it should be possible to model multiple sample data sources concurrently, and then combine these into more robust geostatistical models. It may also be possible to independently verify exposures by multi-parameter co-kriging [17], for example, with laboratory measurements of white blood cell counts in the same samples- such measurements would be expected to be inversely related to radiation dose. Timely and accurate triage assessment will be needed to inform health professionals about individuals at high-risk for Acute Radiation Syndrome, preferably during the prodromal phase. The radiation dose maps generated by the proposed method can potentially contribute to expediting such decisions.

## Supporting information

**S1 Fig. Kriging methods using random points representing 1% of Boston population.** These plumes were generated with random points representing 1% of the Boston (N = 617,594) and Cambridge (N = 105,162) populations, the two subdivisions overlapping the Boston HPAC plume (no precipitation). These points were assigned dose values based on their location relative to the HPAC plume (N = 223 with dose > 0Gy). The following kriging methods were then applied to these data: Ordinary, Simple, Universal and Empirical Bayesian kriging (EBK). A contiguous plume could not be derived from Simple kriging. In these, and in other similar tests, the plume generated using EBK best resembled the plume produced by HPAC.
(TIF)

**S2 Fig. Rural New York scenario at a 90% and 99% stringency of overlap between consecutive iterations of geostatistical analysis.** These plumes represent the radiation levels of the rural New York nuclear incident scenario. The first two plumes were derived using the geostatistical method using two iteration stopping criteria: a 90% (left plume) and a 99% (middle plume) stringency of overlap threshold between consecutive iterations. When compared to the HPAC plume (right

plume), we found that increasing the stringency of overlap resulted in an additional 73 sampling locations selected by densification, which consequently significantly increased the size and similarity of the derived plume, most notably at the 1Gy and 2Gy contours.
(TIF)

**S3 Fig. Boston MA scenario at a 90% and 99% stringency of overlap between consecutive iterations of geostatistical analysis.** These plumes represent the radiation levels of the Boston MA nuclear incident scenario. The first two plumes were derived using the geostatistical method using two different stopping criteria: a 90% (left plume) and a 99% (middle plume) stringency of overlap between consecutive iterations. Right-most plume is HPAC. We find that the increased sampling of this scenario (43 additional sampling locations) led to the development of a gap in the 2Gy threshold, which resulted in a slight decrease in plume accuracy at the 2Gy threshold.
(TIF)

**S1 Table. Comparison between derived and HPAC plumes in terms of area and population.**
(XLSX)

**S2 Table. Comparison between scenario replicates in terms of area.**
(XLSX)

**S3 Table. Radiation scenarios with dose measurement error.**
(DOCX)

**S4 Table. Samples required for plume convergence is inversely related to population density.**
(DOCX)

**S5 Table. Plume derivation with an increased (>99%) stringency of overlap between consecutive iterations of geostatistical analysis.**
(DOCX)

**S1 Methods.**
(DOCX)

## Acknowledgments

We are grateful to the SOSCIP Smart Computing for Innovation Consortium (J.H.M.K., P.K. R.), Natural Sciences and Engineering Research Council of Canada (Engage Program; E.W., P. K.R), Ontario Centres of Excellence (Talent Edge Postdoctoral Fellowship Program; J.H.M.K., P.K.R.), and CytoGnomix (P.K.R., B.C.S, J.H.M.K) for support. We are grateful to Dr. Ruth C Wilkins (Health Canada) for valuable comments on this manuscript.

## Author Contributions

**Conceptualization:** Peter K. Rogan.

**Data curation:** Eliseos J. Mucaki, Ruipeng Lu, Edward Waller.

**Formal analysis:** Peter K. Rogan, Eliseos J. Mucaki, Ruipeng Lu.

**Funding acquisition:** Peter K. Rogan, Joan H. M. Knoll.

**Investigation:** Peter K. Rogan, Eliseos J. Mucaki, Ruipeng Lu.

**Methodology:** Peter K. Rogan, Eliseos J. Mucaki, Ruipeng Lu.

**Project administration:** Peter K. Rogan, Joan H. M. Knoll.

**Resources:** Edward Waller.

**Software:** Ruipeng Lu, Ben C. Shirley.

**Supervision:** Peter K. Rogan, Joan H. M. Knoll.

**Validation:** Peter K. Rogan, Eliseos J. Mucaki, Ruipeng Lu.

**Visualization:** Eliseos J. Mucaki, Ruipeng Lu.

**Writing – original draft:** Peter K. Rogan, Eliseos J. Mucaki, Joan H. M. Knoll.

**Writing – review & editing:** Peter K. Rogan, Joan H. M. Knoll.

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
