## [Decision Letter · Decision Letter 0]

11 Nov 2019

PONE-D-19-22325

Meeting cytogenetic biodosimetry capacity requirements of population-scale radiation exposures with geostatistical sampling

PLOS ONE

Dear Dr. Rogan:

Thank you for submitting your manuscript to PLOS ONE. After careful consideration, we feel that it has merit but does not fully meet PLOS ONE’s publication criteria as it currently stands. Therefore, we invite you to submit a revised version of the manuscript that addresses the points raised during the review process.

Both reviewers noted major revisions for this work with specific comments below.  Please address these in the next submission of the manuscript.

We would appreciate receiving your revised manuscript by Dec 26 2019 11:59PM. To enhance the reproducibility of your results, we recommend that if applicable you deposit your laboratory protocols in protocols.io, where a protocol can be assigned its own identifier (DOI) such that it can be cited independently in the future. For instructions see: http://journals.plos.org/plosone/s/submission-guidelines#loc-laboratory-protocols

We look forward to receiving your revised manuscript.

Kind regards,

Gayle E. Woloschak, PhD

Academic Editor

PLOS ONE

Additional Editor Comments (if provided):

The reviewers have outlined some major concerns that should be addressed in a revision.

2) Thank you for stating the following financial disclosure:

 [We are grateful to the SOSCIP Smart Computing for Innovation Consortium (J.H.M.K.,

P.K.R.), Natural Sciences and Engineering Research Council of Canada (Engage

Program; E.W., P.K.R), Ontario Centres of Excellence (Talent Edge Postdoctoral

Fellowship Program; J.H.M.K., P.K.R.), CytoGnomix (P.K.R.) for support of this project.

We are grateful to Dr. Ruth C Wilkins (Health Canada) for valuable comments on this

manuscript.].               

3) Thank you for stating the following in the Competing Interests section:

[Peter K. Rogan and Joan H. Knoll are founders of CytoGnomix, which partially

supported this research. Cytognomix holds relevant patents and relevant patent

applications.].

4) We note that [Figure(s) #] in your submission contain [map/satellite] images which may be copyrighted. All PLOS content is published under the Creative Commons Attribution License (CC BY 4.0), which means that the manuscript, images, and Supporting Information files will be freely available online, and any third party is permitted to access, download, copy, distribute, and use these materials in any way, even commercially, with proper attribution. For these reasons, we cannot publish previously copyrighted maps or satellite images created using proprietary data, such as Google software (Google Maps, Street View, and Earth). For more information, see our copyright guidelines: http://journals.plos.org/plosone/s/licenses-and-copyright.

a.    You may seek permission from the original copyright holder of Figure(s) [#] to publish the content specifically under the CC BY 4.0 license.

Reviewers' comments:

Reviewer's Responses to Questions

**Comments to the Author**

1. Is the manuscript technically sound, and do the data support the conclusions?

Reviewer #1: Partly

Reviewer #2: Yes

2. Has the statistical analysis been performed appropriately and rigorously? 

Reviewer #1: N/A

Reviewer #2: Yes

3. Have the authors made all data underlying the findings in their manuscript fully available?

Reviewer #1: No

Reviewer #2: Yes

4. Is the manuscript presented in an intelligible fashion and written in standard English?

Reviewer #1: Yes

Reviewer #2: Yes

5. Review Comments to the Author

Reviewer #1: This manuscript has two main elements – one good and one bad. First, the methodology of geostatistical sampling and guidance of future, iterative sampling is logical and well-described. It could have great utility in a scenario of limited data acquisition and processing to replace universal assessment (impossible) with a plan to most rapidly and efficiently acquire the necessary limited data to refine (and confirm) a priori models. However, the application of this approach to the second element – biodosimetry via cytogenetic assessment of dicentric chromosomes – appears to be without merit or value. Also, it is unnecessary. For all the work, the authors have simulated biodosimetry data from physical dose values, with the assumption that the latter is the ground truth. Essentially, this means they could have removed the entire aspect of biodosimetry and simply applied the geostatistical sampling approach to the question of where to focus a limited number of physical dose assessments so as to confirm/modify predicted models of the radiation plume. So they have basically made a circular argument to go from physical dose to biological dose to arrive at agreement with models of physical dose. A second major problem with the application to biodosimetry deals with the re-iterative process. Identification of sites/individuals for subsequent sampling works when the sampling is not a bottleneck – both in terms of capacity (number that can be processed at any time) and throughput (time for generation of the next set of data). The first is obvious and pointed out by the authors – with examples of 160 to 400 samples being presented as possible realistic goal for a “round” of assessments. From this, one-two rounds of testing would be required to provide the necessary samples (Table 1). But, one does not know at start where to sample for each iteration. And one cannot start subsequent sampling until the initial model is produced. With a turn-around time of 1.5 weeks, a model requiring 3-4 iterations would be worthless in terms of timely provision of information on how to triage and who to treat. Even if all bottlenecks are not consecutive, one still has minimum of a week from sampling to a data point (culturing, preparing metaphase cells, and imaging) before the next round can start. Further, if the situation of dose accumulation is not static – which the authors admit it is not – then the data from samples collected in subsequent iterations may not really extend the data in the first iteration. This makes the whole application unfeasible. In contrast, sticking to physical dose assessments, the turn-around time between iterations might be only one day and thus a feasible approach to rapid and efficient real dose-mapping to confirm models.

There is also a problem with the language used in several places. While the whole concept is one of limited sampling to aid in prediction modeling, there are statements that imply universal assessment of all potentially irradiated individuals. Last sentence of first paragraph on page 17 states that a capacity of 400 samples “should be sufficient to diagnose most or all individuals with symptoms of Acute Radiation Syndrome”. First, this has nothing to do with diagnosis, only assignment of a biological dose (>2 Gy) that carries with it a potential for ARS. And with actual symptoms of ARS (a diagnosis) one probably needs nothing further to initiate treatment. Further, the final prediction of biological dose locations (contours) at best will confirm a physical dose map derived from either actual dose measurements or the HPAC models. But what if it doesn’t? Either because there is not a one-to-one correlation between biological dose and physical dose (again, the authors acknowledge this) or because the model is not applicable. And the more “discrepancies” one has in the first iteration, the harder it will be to arrive at the final mapping.

In the end, the use of geostatistical sampling, at least when applied to limited biodosimetry data, is unlikely to be “one of the only practical solutions in a radiation mass casualty to quickly and accurately triage large populations for therapeutic decisions”. It will not be quick – in time to initiate treatment for those who might potentially benefit. And there will be no way to determine how accurate it is.

Reviewer #2: This is an interesting exercise on an attempt to map radiation plumes and fallout to determine a sample of individuals to test for radiation exposure in the case of a nuclear event. There are several concerns around this manuscript.

1. it is not clear how much better this will be compared to simple dosimeters that are commonly found in cities. mapping this mathematically does not appear to add much to the data that is possible to be collected by physical biodosimetry.

2. the major concern is that the radiation doses in a large city will be inhomogeneous due to partial shielding. How does this algorithm account for such events?

6. PLOS authors have the option to publish the peer review history of their article (what does this mean?). If published, this will include your full peer review and any attached files.

Reviewer #1: No

Reviewer #2: No

---

## [Author Response · Author response to Decision Letter 0]

13 Dec 2019

Gayle E. Woloschak, PhD

Academic Editor

PLOS ONE

Dear Dr. Woloschak:

Thank you for allow us to revise our manuscript for PLoS ONE. We appreciate your consideration of the revised manuscript. We are grateful for the insightful comments by the Referees, which we have address in the revision and respond to below.

Please note that we have added one author, Ben C. Shirley, who has contributed software development expertise to this work. 

Kind regards,

Peter K. Rogan, Ph.D. (on behalf of all of the authors)

University of Western Ontario

Responses to Academic Editor

Style requirements: The manuscript is compliant with journal requirements

Financial disclosure: The funders had no role in the design, collection of data, publication or preparation of the manuscript. This is now explicitly stated:

“The funders had no role in study design, data collection and analysis, decision to publish, or preparation of the manuscript."

(please add this text to the original text)

Competing interests:

The text has been revised to:

“Peter K. Rogan and Joan H. Knoll are founders and Ben Shirley is an employee of CytoGnomix, which partially supported this research. Cytognomix holds relevant patents and relevant patent applications. This does not alter our adherence to PLOS ONE policies on sharing data and materials.”

(please replace original text with the above)

Copyrighted images:

All copyrighted map images in the submitted version have been replaced with maps (layer files) downloaded from the publicly available Humanitarian Open Street Map website, which is indicated in the Methods and Figure legends, as requested. These maps span the same latitude and longitude coordinates as the original maps provided by the ArcMap software. This data source is completely open for all uses, and meets the terms of the CC BY 4.0 license.

Response to Reviewers

Comments to the Author

1. Is the manuscript technically sound, and do the data support the conclusions?

Reviewer #1: Partly

Reviewer #2: Yes

Since Reviewer #1 did not indicate specific issues, we are unable to address their response. 

3. Have the authors made all data underlying the findings in their manuscript fully available?

The PLOS Data policy requires authors to make all data underlying the findings described in their manuscript fully available without restriction…

Reviewer #1: No

Reviewer #2: Yes

In response to reviewer #1, we have uploaded the HPAC derived data and our scenario sample locations with inferred radiation levels for each scenario to a Zenodo.archive (https://doi.org/10.5281/zenodo.3572574). The address of this archive is also indicated in the manuscript.

5. Review Comments to the Author

Reviewer #1: 

A) This manuscript has two main elements – one good and one bad. First, the methodology of geostatistical sampling and guidance of future, iterative sampling is logical and well-described. It could have great utility in a scenario of limited data acquisition and processing to replace universal assessment (impossible) with a plan to most rapidly and efficiently acquire the necessary limited data to refine (and confirm) a priori models. However, the application of this approach to the second element – biodosimetry via cytogenetic assessment of dicentric chromosomes – appears to be without merit or value. Also, it is unnecessary. For all the work, the authors have simulated biodosimetry data from physical dose values, with the assumption that the latter is the ground truth. Essentially, this means they could have removed the entire aspect of biodosimetry and simply applied the geostatistical sampling approach to the question of where to focus a limited number of physical dose assessments so as to confirm/modify predicted models of the radiation plume. So they have basically made a circular argument to go from physical dose to biological dose to arrive at agreement with models of physical dose. 

We have modified the text of the manuscript in order to generalize the source of radiation measurement to either physical or biodosimetry methods. While the method was conceptualized with biodosimetry in mind, the method would still be useful to reduce physical sampling requirements in population-scale radiation scenarios and would be expected to decrease total radiation exposures of first responders. We have generalized the text to avoid specifying a particular dosimetry method (e.g. “cytogenetic biodosimetry” changed to “dosimetry”). As this transition required numerous edits (too many to describe here), we have provided select examples of where the text has been modified to include physical dose measurement:

1) Abstract (Background section): 

“Even with high-throughput dosimetry analysis, test volumes far exceed the capacities of first responders to measure radiation exposures directly, or to acquire and process samples for follow-on biodosimetry testing.”

2) Abstract (Results/Conclusions section): “Geostatistical mapping limits the number of individuals requiring laboratory testing, the time required, and radiation exposure to first responders.”

This has been modified to (see bolded text):

“Geostatistical mapping limits the number of individuals requiring dose assessment, the time required, and radiation exposure to first responders.”

3) Introduction (Paragraph 4, page 4): “The question this paper addresses is whether the radiation plumes derived by HPAC can be reconstructed with kriging using a relatively small number of samples analyzed by physical or biodosimetry methods.”

4) Results (Paragraph 1, page 11): “We simulated the analysis of 28 population-scale, 10 megaton yield nuclear detonation scenarios with ADCI-HT using cytogenetic biodosimetry data…”

This has been modified to:

 “We simulated the analysis of 28 population-scale, 10 megaton yield nuclear detonation scenarios with dosimetry data…”

5) We have also greatly modified the Introduction paragraph describing dosimetry methods to expand information regarding physical dose methods. Example:

Introduction (Paragraph 2, page 3): “Physical dose can be inferred from the prodromal response of absorbed radiation, which include erythema, headache, fever, lethargy, tachycardia, nausea and vomiting (1,3).”

However, results from biodosimetry can be achieved in a clinically relevant time frame. The estimates given in the manuscript were simply examples based on certain assumptions about laboratory capacity. With laboratory scale-up and parallelization (e.g. commercial cytogenetic laboratories with highly automated sample preparation and redundant microscope imaging systems), the time to determine and report absorbed radiation would be reduced significantly, e.g. 3-4 days. This approach has been suggested as part of the U.S. Concept of Operations in a nuclear incident, which is now cited in this article as reference 36. We previously suggested this possibility at a BARDA TechWatch presentation in 2012 and provided relevant data to these authors (who presented it at the most recent EPR Biodose conference in Munich, 2018). As the text has been modified to include physical dosimetry methods, the majority of text discussing cytogenetic biodosimetry has been moved to the latter part of the Discussion (pages 18-20). 

We carefully describe the assignment of dose values in the simulation to demonstrate that the results presented are not the result of cyclical reasoning. The HPAC plume is considered the “ground truth” dose estimate, however the actual portable data generated and exported by the HPAC software is confined to the values at the contours that make up the plume itself. Locations between the geospatial coordinates given by the contours are not provided by HPAC. For more accurate dose estimation, these values should be interpolated between adjacent contours. To avoid a circular reasoning argument we have not performed such interpolation, rather we use the dose estimate of the outer contour adjacent to sampled locations (regardless of whether they intersect the contour or not), that is, we introduce a source of systematic error to avoid the circular argument. The data collected by first responders after kriging and densification should reflect the actual dose at the site sampled. When we then compare the derived plume with the HPAC plume, we are comparing the dose that was derived from the corresponding interval separated by contours with what is expected from geostatistical estimation of dose. The goal was to demonstrate that this approximation method provides actionable information about exposures based on the sampling procedure, and that we can reconstruct the plume from a small fraction of the samples that were used to generate it (<10%). Biological dose estimation for large scale events with geolocation data equivalent to HPAC models has not been previously reported. Without such a map, it is not possible to directly compare bio- with physical dose estimates for a large-scale incident. We therefore simulate the biological dose by sampling the HPAC map at many fewer locations that HPAC uses to derive plume contours.

B) A second major problem with the application to biodosimetry deals with the re-iterative process. Identification of sites/individuals for subsequent sampling works when the sampling is not a bottleneck – both in terms of capacity (number that can be processed at any time) and throughput (time for generation of the next set of data). The first is obvious and pointed out by the authors – with examples of 160 to 400 samples being presented as possible realistic goal for a “round” of assessments. From this, one-two rounds of testing would be required to provide the necessary samples (Table 1). But, one does not know at start where to sample for each iteration. And one cannot start subsequent sampling until the initial model is produced. With a turn-around time of 1.5 weeks, a model requiring 3-4 iterations would be worthless in terms of timely provision of information on how to triage and who to treat. Even if all bottlenecks are not consecutive, one still has minimum of a week from sampling to a data point (culturing, preparing metaphase cells, and imaging) before the next round can start. Further, if the situation of dose accumulation is not static – which the authors admit it is not – then the data from samples collected in subsequent iterations may not really extend the data in the first iteration. This makes the whole application unfeasible. In contrast, sticking to physical dose assessments, the turn-around time between iterations might be only one day and thus a feasible approach to rapid and efficient real dose-mapping to confirm models.

We have added text to address where first-responders could perform initial sampling (Methods section, first paragraph, page 4):

“Initially, locations within likely fallout areas were sampled from the potentially affected population. In a real-world scenario, initial randomly sampled locations would be selected based on approximate wind direction and location of the epicenter (“ground zero”).”

We have also made significant changes throughout the text in order to generalize the potential sources of radiation measurements to include physical dose. The methodology proposed in this manuscript should be feasible regardless of the dosimetry method used to derive it. For any radiation metric, the proposed approach should benefit first responders by reducing sampling requirements, and consequently their exposure to radiation.

Regarding your comments about cytogenetic biodosimetry turnaround time, please see our comment in part (A).

C) There is also a problem with the language used in several places. While the whole concept is one of limited sampling to aid in prediction modeling, there are statements that imply universal assessment of all potentially irradiated individuals. Last sentence of first paragraph on page 17 states that a capacity of 400 samples “should be sufficient to diagnose most or all individuals with symptoms of Acute Radiation Syndrome”. First, this has nothing to do with diagnosis, only assignment of a biological dose (>2 Gy) that carries with it a potential for ARS. And with actual symptoms of ARS (a diagnosis) one probably needs nothing further to initiate treatment. 

Thank you for your suggestion. We have corrected any text that describes Acute Radiation Syndrome (ARS) to make it clear that a >2Gy dose only indicates the potential risk of ARS and is not a diagnosis of ARS. For example, the sentence you indicated on page 17 (now on page 19) is now written as follows:

“Timely and accurate assessment will be needed to inform health professionals about individuals at high-risk for Acute Radiation Syndrome, preferably during the prodromal phase.”

Another example where we discussing a method to expedite identification of those at risk for ARS, not diagnosis of ARS:

Introduction (paragraph 1, page 3): “Individuals at risk for acute radiation syndrome, e.g. those receiving ≥ 2Gy exposure, should be rapidly identified by these approaches to determine eligibility for therapy (2).”

D) Further, the final prediction of biological dose locations (contours) at best will confirm a physical dose map derived from either actual dose measurements or the HPAC models. But what if it doesn’t? Either because there is not a one-to-one correlation between biological dose and physical dose (again, the authors acknowledge this) or because the model is not applicable. And the more “discrepancies” one has in the first iteration, the harder it will be to arrive at the final mapping.

The manuscript already addresses this issue. We have explicitly introduced the possibility that the deviation could involve physical vs biodosimetry dose estimation:

Results (final paragraph, page 16): “Maximum deviations were designed to represent confidence values in physical and/or biodosimetry methods, including dicentric analysis (DCA; ±0.5Gy) or cytokinesis-blocked micronucleus assays (±1.0Gy), which exhibit higher variance in estimating dose compared to DCA (28–30). The introduction of such measurement errors led to plumes that were generally less accurate, required a greater number of samples, and additional iterations of kriging and densification to achieve convergence (S2 Table). The degree of error in this analysis was also correlated with decreased accuracy and increased processing time (±0.5Gy error-derived plumes were more accurate than ±1.0Gy error-derived plumes in nearly all scenarios; S2 Table).”

Note that while an increase in dosimetry error both increases sampling requirements and modestly decreases overall accuracy, the method will still ultimately derive a plume regardless. 

E) In the end, the use of geostatistical sampling, at least when applied to limited biodosimetry data, is unlikely to be “one of the only practical solutions in a radiation mass casualty to quickly and accurately triage large populations for therapeutic decisions”. It will not be quick – in time to initiate treatment for those who might potentially benefit. And there will be no way to determine how accurate it is.

We have altered this statement:

Discussion (paragraph 7; page 20): “Geostatistical sampling may provide a practical solution to quickly triage these exposures in large populations.”

There are promising approaches to expedite the preparation of samples and data acquisition steps (Dainiak et al. 2019; Ref. #36) so that triage biodosimetry could be carried out sufficiently quickly to benefit many of those requiring treatment. Furthermore, we have broadened the scope of the paper to include reconstruction of physical radiation plumes from geostatistically sampled subsets of locations or individuals. Reconstruction of the plume could be carried out within a few minutes if the radiation detectors are placed at geolocations specified by kriging and densification. 

Reviewer #2: 

This is an interesting exercise on an attempt to map radiation plumes and fallout to determine a sample of individuals to test for radiation exposure in the case of a nuclear event. There are several concerns around this manuscript.

1. it is not clear how much better this will be compared to simple dosimeters that are commonly found in cities. mapping this mathematically does not appear to add much to the data that is possible to be collected by physical biodosimetry.

In the Introduction, we cite previous studies that illustrate inaccuracies when comparing results of physical dosimetry and cytogenetic biodosimetry for the same individuals from the same exposure. Such discrepancies have implications for medical emergency management: 

Introduction (Paragraph 2, page 3): “Confounding factors may affect testing results. These include the possibility of discordant physical radiation measurements and estimated biodosimetry exposures (13,14), and other causes of indirect predictors of exposure such as time-to-emesis (such as head trauma, or comorbid infections), thereby reducing accuracy of diagnoses (15).”

The manuscript has been modified to put a greater emphasis on the benefits of this method for reconstructing exposures over affected geographic areas for any type of dosimetry, including physical dosimetry (see our response to Reviewer #1 [A]). Despite physical dosimetry being much faster compared to cytogenetic biodosimetry, determining the physical dose of all individuals in a large city is still infeasible. In the early stages of a nuclear event, collecting data would be difficult for first responders. In this situation, the proposed approach would allow for the development of an accurate radiation plume with only limited sampling while decreasing exposure of first responders by minimizing time in irradiated areas.

2. the major concern is that the radiation doses in a large city will be inhomogeneous due to partial shielding. How does this algorithm account for such events?

The third paragraph of the Discussion describes the limitations of our method. Here, we discuss how shielding (or “shadowing”) by infrastructure is not accounted for, and how these concessions may limit the overall accuracy of our method. 

Results (Paragraph 3, page 18): “Finally, neither the HPAC version available for this study, nor our geostatistical models account for shielding by infrastructure, such as shadowing, which computes the degree to which radiation is prevented from reaching certain locations by the urban environment (34). These concessions may limit the overall accuracy of radiation measurements that we are using as the ground truth.”

---

## [Decision Letter · Decision Letter 1]

30 Dec 2019

PONE-D-19-22325R1

Meeting radiation dosimetry capacity requirements of population-scale exposures by geostatistical sampling

PLOS ONE

Dear Dr. Rogan:

Thank you for submitting your manuscript to PLOS ONE. After careful consideration, we feel that it has merit but does not fully meet PLOS ONE’s publication criteria as it currently stands. Therefore, we invite you to submit a revised version of the manuscript that addresses the points raised during the review process.

Please address the concerns raised by the reviewers.  One suggested major revisions.

We would appreciate receiving your revised manuscript by Feb 13 2020 11:59PM. To enhance the reproducibility of your results, we recommend that if applicable you deposit your laboratory protocols in protocols.io, where a protocol can be assigned its own identifier (DOI) such that it can be cited independently in the future. For instructions see: http://journals.plos.org/plosone/s/submission-guidelines#loc-laboratory-protocols

We look forward to receiving your revised manuscript.

Kind regards,

Gayle E. Woloschak, PhD

Academic Editor

PLOS ONE

Additional Editor Comments (if provided):

One reviewer accepted the work, the other suggested major revisions. Please address as many concerns as possible.

Reviewers' comments:

Reviewer's Responses to Questions

**Comments to the Author**

1. If the authors have adequately addressed your comments raised in a previous round of review and you feel that this manuscript is now acceptable for publication, you may indicate that here to bypass the “Comments to the Author” section, enter your conflict of interest statement in the “Confidential to Editor” section, and submit your "Accept" recommendation.

Reviewer #1: (No Response)

Reviewer #2: All comments have been addressed

2. Is the manuscript technically sound, and do the data support the conclusions?

Reviewer #1: Partly

Reviewer #2: Yes

3. Has the statistical analysis been performed appropriately and rigorously? 

Reviewer #1: No

Reviewer #2: Yes

4. Have the authors made all data underlying the findings in their manuscript fully available?

Reviewer #1: Yes

Reviewer #2: Yes

5. Is the manuscript presented in an intelligible fashion and written in standard English?

Reviewer #1: Yes

Reviewer #2: No

6. Review Comments to the Author

Reviewer #1: While substantial improvements have been made, there is still a remaining problem that will mislead a naïve reader. That is the continued emphasis on ”individuals” that implies more than is warranted. While the authors have made the necessary modifications to replace the biological dosimetry with physical dose measurement, there is still text that implies assessment of individuals (i.e., biological dosimetry) rather than locations (physical dosimetry). Although the simulated measurements based on location can be weighted for population density, it still does not provide information at the individual level.

An example of the focus on individuals is a phrase that is repeated in the Abstract (Results) and the first paragraph of the Discussion:

On average, 71±9% of those with ≥2Gy exposures were accurately localized.

plumes from 28 distinct scenarios simulating absorbed radiation identified 71±9% of individuals with ≥2Gy exposures

A better phrasing would be “71±9% of those locations where the population might be expected to receive an exposure greater than 2 Gy were identified.”

There is also a problem with how the central data in Table 1 are interpreted. As stated on page 13, “Success” was based on accuracy of predicting irradiated samples greater than 2 Gy. The authors took the final iteration values and stated an average accuracy of 70.6%. But this is after omitting an outlier for Columbia (or replacing with a further re-iteration attempt – it is not clear which). There was also an intermediate outlier for Columbus but the final value was in range and was apparently used. No mention is made of Charleston where the final value exceeds the 10% threshold of difference from either of the first two values. Most importantly, the average accuracy from the final values after the process is no different from the average accuracy computed using the first iteration values. Just as many locations had a decrease from first to final iteration as had an increase. And excluding the 10% increase for Charleston the maximum “improvement” was 4% while the maximum loss of accuracy was 5%. So, what is the value of this iterative process? While it may have the potential to refine predicted dose maps, it clearly does not provide better results for “success” based on the accuracy values.

Finally, despite the shift from biodosimetry to physical dose sampling, there is still the underlying implication that this process may be applied to “a relatively small number of samples analyzed by physical or biodosimetry methods” (last sentence of Introduction). Similarly, on page 20, “Population-scale radiation exposure identification can be achieved through a combination of high-throughput dicentric chromosome identification software and GIS-based software analysis, and the test volume is likely to be feasible for a large dosimetry lab.” Any aspect that relies on an iterative process using cytogenetic data is patently not feasible for triage purposes when there is a minimum 1 week turn-around time. One might use the reiterative process on physical dose mapping to select locations where selected individuals are then assessed by biodosimetry – but this is not the take-away message from the manuscript as written.

Reviewer #2: all concerns addressed - no further comments

7. PLOS authors have the option to publish the peer review history of their article (what does this mean?). If published, this will include your full peer review and any attached files.

Reviewer #1: No

Reviewer #2: No

---

## [Author Response · Author response to Decision Letter 1]

10 Jan 2020

Academic Editor Comments 

We would appreciate receiving your revised manuscript by Feb 13 2020 11:59PM. To enhance the reproducibility of your results, we recommend that if applicable you deposit your laboratory protocols in protocols.io, where a protocol can be assigned its own identifier (DOI) such that it can be cited independently in the future. For instructions see: http://journals.plos.org/plosone/s/submission-guidelines#loc-laboratory-protocols

Author Response (Cover letter)

Dear Dr. Woloschak:

Thank you for allowing us to once again revise our manuscript for PLoS ONE. We have thoughtfully considered the comments made by the first referee, and we have responded to each point in a separate letter provided below. As you recommended, we have also included a detailed protocol for generation and comparison of radiation dosimetry maps using the method described in the paper and uploaded this protocol to protocols.io (http://dx.doi.org/10.17504/protocols.io.ba4nigve). This protocol has been named "Protocol for Geostatistical Determination of Radiation Dosimetry Maps of Population-Scale Exposures".

The first referee expresses their specific point of view regarding the protocols and guidelines for application of physical dosimetry vs biological dosimetry methods and relegates biological dosimetry to a secondary role. Recent advances in these techniques, in particular interpretation of data, have demonstrated that timely and useful information can inform clinical decisions. We never intended our manuscript to be a discourse on benefits vs. drawbacks of performing different types of dosimetry. Discussion of preference between physical and bio-dosimetry methods is not relevant to the scientific validity of the work, which is one of the reasons why we submitted our manuscript to PLoS ONE. 

The point of our manuscript is to introduce a new method of deriving accurate radiation plumes requiring far fewer direct measurements than previously thought necessary. While plumes can be derived with the geostatistical approach described in our manuscript using any dosimetry method, it has particular implications for biodosimetry, which has been previously dismissed by many (including this reviewer) as too slow to be used to guide management of radiation-caused illness. Rather, maps derived from biological radiation measurements that significantly exceeding the recognized threshold of 2Gy are likely precise enough to begin treatment prior to obtaining the final plume. 

Putting the issue of the benefits or drawbacks of performing different types of dosimetry aside, we hope that you find our responses to the first reviewer’s comments acceptable; and that you consider our manuscript for publication in PLoS ONE. 

Kind regards,

Peter K. Rogan, Ph.D. (on behalf of all of the authors)

Reviewer comments (Author Responses follow each comment)

1. If the authors have adequately addressed your comments raised in a previous round of review and you feel that this manuscript is now acceptable for publication, you may indicate that here to bypass the “Comments to the Author” section, enter your conflict of interest statement in the “Confidential to Editor” section, and submit your "Accept" recommendation.

Reviewer #1: (No Response)

Reviewer #2: All comments have been addressed

2. Is the manuscript technically sound, and do the data support the conclusions?

Reviewer #1: Partly

Reviewer #2: Yes

RESPONSE 

Reviewer #1 has misunderstood the results by confusing the “iterative” process of kriging and densification with the analysis of 3 completely independent replicates of scenarios initiated using different random seed conditions (see Section 6.3 below). We have submitted a detailed protocol to the protocols.io website that demonstrates this iterative process. We have also uploaded the sample data for each replicate of each scenario, along with project-related software that are used with ArcMap to perform the geostatistical analyses to the public repository, Zenodo. Reviewer #1 is invited to review these resources and reproduce our findings prior to drawing conclusions regarding their adequacy.

3. Has the statistical analysis been performed appropriately and rigorously?

Reviewer #1: No

Reviewer #2: Yes

RESPONSE 

There is no factual basis for this conclusion by Reviewer #1. We emphatically deny that we have removed or selectively altered any data presented in this study. In fact, the reviewer’s conclusion appears to have been based on a misinterpretation of the results obtained for the Columbia SC scenario (see Section 6.3 below) alone. The statistical analyses for all 24 scenarios have been performed appropriately and rigorously, according to accepted methods in geostatistical analyses. 

As mentioned in our previous response, we have uploaded the HPAC-derived data and our scenario sample locations with inferred radiation levels for each scenario to a Zenodo archive (https://doi.org/10.5281/zenodo.3572574). In addition to this, we also include a detailed protocol for generation and comparison of radiation dosimetry maps using the method described in the paper and uploaded this protocol to protocols.io (dx.doi.org/10.17504/protocols.io.ba4nigve). The protocol contain links to data and programs in the Zenodo archive, however these links will not be live until the archive is formally published. This will occur should the paper be acceptable for publication by PLoS ONE.

4. Have the authors made all data underlying the findings in their manuscript fully available?

Reviewer #1: Yes

Reviewer #2: Yes

5. Is the manuscript presented in an intelligible fashion and written in standard English?

Reviewer #1: Yes

Reviewer #2: No

RESPONSE 

We do not understand this conclusion by reviewer 2, who has recommended acceptance of the manuscript. Elaboration is requested.

6. Review Comments to the Author

Reviewer #1: 

1. While substantial improvements have been made, there is still a remaining problem that will mislead a naïve reader. That is the continued emphasis on ”individuals” that implies more than is warranted. While the authors have made the necessary modifications to replace the biological dosimetry with physical dose measurement, there is still text that implies assessment of individuals (i.e., biological dosimetry) rather than locations (physical dosimetry). Although the simulated measurements based on location can be weighted for population density, it still does not provide information at the individual level.

RESPONSE

The method described in this manuscript would allow for the derivation of a radiation plume using any (consistent) type of radiological measurement. This includes the physical measurement of radiation of a location, but this also includes biological dosimetry of individuals, which is what we are referring to here. The Introduction describes how first responders and medical personnel have qualitatively inferred exposures based upon presentation of symptoms, aside from simply obtaining direct measurements via Geiger counters:

“Physical dose can be inferred from the prodromal response of absorbed radiation, which include erythema, headache, fever, lethargy, tachycardia, nausea and vomiting (1,3).”

To eliminate any confusion, we have added text to the method which indicates that “sample” can refer to any radiological measurement type:

Methods: “…we refer to these locations as samples, regardless of whether the radiation is quantified from either physical emissions or from absorbed biological effects. We assume that all measurements used to derive the plume are consistently obtained by the same approach.”

Biodosimetry of individuals can provide actual exposure measurements, whereas physical dosimeters measure environmental radiation emission levels. While physical dosimetry measurement is more rapid than biodosimetry, there are limitations in applying unfiltered data to derivation of radiation plumes. There are well known false positive readouts obtained through aerial physical dosimetry counterterrorism surveys (Karam PA, “Radiation in Daily Life,” ANS Nuclear Cafe website; Aug. 28, 2017):

http://ansnuclearcafe.org/2017/08/28/radiation-in-daily-life/#sthash.pSWLh6N3.dpbs)

Common environmental sources of radiation, such as large deposits of granite (i.e. headstones in cemeteries), as well as medical and construction sites (i.e. radiation from poorly maintained X-Ray machines), can lead to false positive spikes in physical radiation unrelated to the nuclear incident that we are attempting to model using multiple measurements dispersed across the landscape. If used, these data could distort a plume derived by geostatistical analysis. These factors would be somewhat mitigated by depositing physical detectors at ground locations specified by this iterative densification, as suggested in the paper. Nevertheless, it would appear that there are certain advantages of performing biodosimetry on individuals at specified locations, as an alternative to surveying locations themselves for radioactive emissions.

2. An example of the focus on individuals is a phrase that is repeated in the Abstract (Results) and the first paragraph of the Discussion:

On average, 71±9% of those with ≥2Gy exposures were accurately localized.

plumes from 28 distinct scenarios simulating absorbed radiation identified 71±9% of individuals with ≥2Gy exposures

A better phrasing would be “71±9% of those locations where the population might be expected to receive an exposure greater than 2 Gy were identified.”

RESPONSE

The value of 71±9% accuracy has been corrected (to 70±10%) in the latest version of the manuscript. As the paper states, each scenario initially consisted of three replicates, however we did perform additional replicates for 2 of the scenarios which exhibited larger inter-replicate variation in performance. An additional 2 replicates for these scenarios were included close to the completion of the manuscript. Inclusion of these additional replicates (Columbia SC replicates #4 and 5, and the Columbus OH replicates #4 and 5) resulted in a minor adjustment to the accuracy calculation, which was not statistically different from the results based on 3 replicates. The omission of the additional replicates from the accuracy calculation was inadvertent.

We appreciate the reviewer’s suggestion, and have made said change to the manuscript, with slight adjustments:

Abstract: “On average, 70±10% of locations where populations are expected to receive an exposure ≥2Gy were identified.”

Discussion: “Overall, plumes from 28 distinct scenarios simulating absorbed radiation identified 70±10% of locations where the population are expected to receive ≥2Gy of radiation exposure (and thus are eligible for cytokine therapy), …”

Furthermore, the overall accuracy of 70.6% has been adjusted to 69.8%.

3. There is also a problem with how the central data in Table 1 are interpreted. As stated on page 13, “Success” was based on accuracy of predicting irradiated samples greater than 2 Gy. The authors took the final iteration values and stated an average accuracy of 70.6%. But this is after omitting an outlier for Columbia (or replacing with a further re-iteration attempt – it is not clear which). 

There was also an intermediate outlier for Columbus but the final value was in range and was apparently used. No mention is made of Charleston where the final value exceeds the 10% threshold of difference from either of the first two values. Most importantly, the average accuracy from the final values after the process is no different from the average accuracy computed using the first iteration values. Just as many locations had a decrease from first to final iteration as had an increase. And excluding the 10% increase for Charleston the maximum “improvement” was 4% while the maximum loss of accuracy was 5%. So, what is the value of this iterative process? While it may have the potential to refine predicted dose maps, it clearly does not provide better results for “success” based on the accuracy values.

RESPONSE

This was a misunderstanding by the reviewer, since outliers were not excluded at any time in any computation. The final accuracy for each of the replicates (replicate #1, 2 and 3) were included in the computation of the overall accuracy of the project. As previously mentioned, inclusion of replicates #4 and 5 for Columbia SC and Columbus OH slightly decreased this value (from 70.6% to 69.8%).

There also appears to have been a fundamental misunderstanding of the results presented in Table 1. In contrast to the assertion by the reviewer, replicates shown in Table 1 are not a step-by-step description of the progression of our iterative process. Instead, they each represent independently-derived plumes (each consisting of multiple iterations of kriging and densification) differing only in their starting set of random determined sample locations. The replicates are arbitrarily labeled as numbers 1,2 and 3. These replicates are not presented in any particular order. 

From the Methods:

“Comparisons between independent replicates of the same scenarios using different, randomized, initial sample distributions evaluated the reliability of this approach.”

“The geostatisical workflow was used to analyze three replicates of each scenario, with each replicate initiated with a different set of random sample locations within the plume.”

The plumes derived from the initial set of points alone are less accurate than the final plume. In low population-density or sparsely sampled regions, the HPAC and derived plumes show limited overlap of the same contours resulting in poor accuracy for defining the 2Gy threshold. Neither example in Figure 2 exhibits this issue because of the relatively higher sampling densities in these scenarios. Reduction in the off-diagonal cell values over consecutive iterations in the tables below the plume maps illustrates the improvement in the model. We do not believe that it would be as informative to the reader to compute accuracy metrics between iterations, and generally only present accuracy at the end of the procedure (except for the new Columbia SC replicate shown below). 

Nevertheless, we address the reviewer’s critique of our results for the Columbia SC scenario by performing this analysis for a new, previously unreported replicate. The following results are now reported as an example of the procedure (at https://www.protocols.io/) described in the Methods of our manuscript.

We find:

• Accuracy of plume from initial set of random samples only: 0% accuracy (the entire plume is <2Gy; image of plume in Section 4 of the online protocol) [N=8 samples > 0 Gy]

• Accuracy after 1 Iteration: 21.1% accuracy (image of plume in Section 5 of the online protocol) [N=55 samples > 0 Gy]

• Accuracy after 3 Iterations: 46.0% accuracy [N=79 samples > 0 Gy]

• Accuracy after 5 Iterations: 68.4% accuracy (final derived plume; image of plume in Section 6 of the online protocol) [N=136 samples > 0 Gy]

As this was a source of confusion, we have added a new comment in the footnote to Table 1 for the “No. of Iterations” column:

“2 The number of iterations (kriging and densification steps) required to reach stopping criteria for this replicate.”

Additionally, in places in the manuscript where we discuss the iterative process, we have added text to make it clear that we are referring to iterations of kriging and densification. For example:

Methods (new text bolded): “These kriging and densification processes are repeated for a limited number of iterations until the coverage area and the radiation level contours of the inferred plume stabilizes”

4. Finally, despite the shift from biodosimetry to physical dose sampling, there is still the underlying implication that this process may be applied to “a relatively small number of samples analyzed by physical or biodosimetry methods” (last sentence of Introduction). Similarly, on page 20, “Population-scale radiation exposure identification can be achieved through a combination of high-throughput dicentric chromosome identification software and GIS-based software analysis, and the test volume is likely to be feasible for a large dosimetry lab.” Any aspect that relies on an iterative process using cytogenetic data is patently not feasible for triage purposes when there is a minimum 1 week turn-around time. One might use the reiterative process on physical dose mapping to select locations where selected individuals are then assessed by biodosimetry – but this is not the take-away message from the manuscript as written.

RESPONSE

We have not “shifted the emphasis” of this paper to sampling by physical dosimetry. Rather, we have expanded our approach to include sample data originating from either physical- or bio-dosimetry. The approach we describe can be performed using either type of data. 

Biological dosimetry would still be feasible to generate meaningful plumes, leading to sufficiently timed treatment. The 1.5-week scenario that we describe in the Discussion accounted for all iterations of the plume development process and assumed that a single or two typical biodosimetry laboratories would perform the sample processing and analysis. The revised manuscript cited a recent Concept of Operations study (Daniak et al. 2019) that envisions engagement by large commercial cytogenetic operations with substantial automation of these processes. We have attempted to make this clearer in the text (new text bolded):

“The expected throughput of cytogenetic biodosimetry of a typical small city scenario (~160 samples) is less than 1 week (for all iterations per scenario, cumulatively), assuming…”

While sample culturing introduces an initial 48-hour lag, the period afterwards would be quite rapid due to A) automated sample preparation, B) multiplexed imaging of samples in parallel, C) automated dicentric chromosome detection and radiation dose estimation, and D) kriging and densification. Additional samples can be collected during some of these analysis steps. We envision a continuous process, not a segmented one as the reviewer has suggested.

Geographic regions close to the epicentre receiving high levels of radiation exposures (> 3Gy) will be evident early in the process. It should not be necessary to wait until the final iteration is completed to apply this information in clinical decision making. We have not investigated this possibility in detail in the current manuscript, however we plan to examine this possibility in future studies. If proven, it may be feasible to consider eligibility for therapy using the plume derived after the first iteration of geostatistical analysis. While the 2Gy contour may not be well defined after a single iteration, we note that locations where absorbed radiation exposures are highest generally cover the similar geographic contours in subsequent plumes. Individuals in imminent need of treatment would benefit sooner, while management of those with lower, clinically relevant radiation exposure would be deferred until the plume is better defined.

It is not our intent to evaluate or compare physical versus biological dosimetry methods in this study. The purpose was to introduce a geostatistical method to deriving radiological plumes from any measurement source. Biodosimetry covers many techniques, however our work has primary focused on advancing the state-of-the-art of the dicentric chromosome assay (see https://adciwiki.cytognomix.com/doku.php?id=main:references). The Discussion of this paper therefore describes the impact of the geostatistical approach on this assay. Other, more rapid biodosimetry assays could also be used to derive a plume based on absorbed radiation, however the dose estimates obtained with these techniques have lower confidence. We also present results addressing the impacts of less accurate measurements on derived radiation plumes, whether they are derived from biological or physical dosimetry data.

---

## [Decision Letter · Decision Letter 2]

28 Jan 2020

PONE-D-19-22325R2

Meeting radiation dosimetry capacity requirements of population-scale exposures by geostatistical sampling

PLOS ONE

Dear Dr. Rogan--

Thank you for submitting your manuscript to PLOS ONE. After careful consideration, we feel that it has merit but does not fully meet PLOS ONE’s publication criteria as it currently stands. Therefore, we invite you to submit a revised version of the manuscript that addresses the points raised during the review process.

One reviewer noted minor revisions, but believes the work is very important as a contribution to the field.  The other reviewer has selected rejection even though in the most recent decision selected acceptance.  Please attempt to address concerns of BOTH reviewers, I will attempt to work through the reviewer who changed opinions on the work so drastically.

We would appreciate receiving your revised manuscript by Mar 13 2020 11:59PM. To enhance the reproducibility of your results, we recommend that if applicable you deposit your laboratory protocols in protocols.io, where a protocol can be assigned its own identifier (DOI) such that it can be cited independently in the future. For instructions see: http://journals.plos.org/plosone/s/submission-guidelines#loc-laboratory-protocols

We look forward to receiving your revised manuscript.

Kind regards,

Gayle E. Woloschak, PhD

Academic Editor

PLOS ONE

Additional Editor Comments (if provided):

One reviewer has indicated that minor revisions are needed. One reviewer has rejected the work, but previously accepted it. Please address as many concerns as you can and I will attempt to wade through the issues of reviewers changing their opinions on the work.

Reviewers' comments:

Reviewer's Responses to Questions

**Comments to the Author**

1. If the authors have adequately addressed your comments raised in a previous round of review and you feel that this manuscript is now acceptable for publication, you may indicate that here to bypass the “Comments to the Author” section, enter your conflict of interest statement in the “Confidential to Editor” section, and submit your "Accept" recommendation.

Reviewer #1: (No Response)

Reviewer #2: All comments have been addressed

2. Is the manuscript technically sound, and do the data support the conclusions?

Reviewer #1: Partly

Reviewer #2: No

3. Has the statistical analysis been performed appropriately and rigorously? 

Reviewer #1: Yes

Reviewer #2: Yes

4. Have the authors made all data underlying the findings in their manuscript fully available?

Reviewer #1: Yes

Reviewer #2: Yes

5. Is the manuscript presented in an intelligible fashion and written in standard English?

Reviewer #1: Yes

Reviewer #2: No

6. Review Comments to the Author

Reviewer #1: This manuscript is definitely getting better; however, there remain problematic issues that relate to the potential for a causal reader to be misled as to what was done and what the results mean.

The Abstract is now an appropriate description of what was done. A minor suggestion is to move the first sentence of the Methods to before the current third sentence (Initially …). This would emphasize that this is a modelling exercise and that exposures or boundaries were not actually determined.

One remaining issue in the manuscript body is in the overlap or inappropriate implied equivalence of “individuals” and “locations”. The authors should confirm (by searching for the words) that when talking about measurements or sampling or results that location is applied to physical dosimetry and individual is restricted to biodosimetry. Some of these may just be residual from prior versions but need to be corrected. This would then allow the appropriate inference by the reader that locations are fixed and may accurately reflect physical dose levels at or in proximity to that location. This then leads to a reasonable assumption that interpolation between two locations (by additional sampling) is likely to refine dose boundaries. The same cannot be said for individuals. A value for a third individual, sampled at a location between the locations where two initial individuals were assessed by biodosimetry, would not a priori be expected to provide an intermediate biodosimetry value. This could be due to heterogeneity in inherent radiosensitivity of the population, movement by the individual, comorbidities, or any of the reasons that the authors allude to. This also leads into the need to be precise as to what measurements are reasonable to utilize for the kriging process and the re-iterative process. That is, one can reasonably postulate additional physical dosimetry sampling over a few days to refine dose boundaries that then can be used to perform biodosimetry on a selected set of individuals. But with the time restraints of biodosimetry it is not practical to consider more than one sampling as contributing to the triaging process. But current wording might suggest otherwise. As an example, at the bottom of page 3, there is the following:

“One approach to alleviating the need to triage all potentially exposed individuals would be to survey a subset of individuals combined with their respective locations by geostatistical analysis. We demonstrate that combining such surveys with the geolocations of these measurements can reduce sampling requirements in population-scale radiation scenarios and would be expected to decrease total radiation exposures of first responders.”

The term “combining” might be interpreted as using the two dosimetry methods reiteratively. Also “surveys” of individuals might be mis-interpreted as multiple surveys (i.e., biodosimetry) in the same general vicinity to aid in the kriging, rather than a single targeted survey of individuals in a location selected by the physical dosimetry kriging.

Last sentence before Methods: “kriging using a relatively small number of samples analyzed by physical or biodosimetry methods.” Biodosimetry needs to be removed, but the sentence can be extended to state using kriging of physical dosimetry to aid in selection of individuals for biodosimetry.

Page 5: “Densification is the geostatistical procedure that targets and localizes an additional small cohort of irradiated individuals (1) to mitigate uncertainty in environmental measurement. These kriging and densification processes are repeated for a limited number of iterations until the coverage area and the radiation level contours of the inferred plume stabilizes (i.e. sampling of additional individuals (2)”. (1) it identifies a location, not any individuals who might be there. (2) Locations will be sampled reiteratively, not individuals.

Page 7, para 2: “To map predicted radiation levels based on the distribution of dose-estimated patients around the prediction location” It is locations sampled by physical dosimetry that have a distribution – not patients.

Figure legend 2; last sentence “% of the individuals eligible for cytokine treatment” – locations (weighted for population density) where treatments of individuals might be necessary.

Page 15; para 2; sentence 1: “subset of irradiated individuals”. Subset of locations - there is no a priori assumption about irradiation of individuals. Sentence 2: unirradiated individuals. Sentence 3: irradiated individuals.

Page 16; para 2; sentence 1: “incorrect sampling of potentially exposed individuals”.

Page 16; para 2; last sentence: “fewer irradiated individuals”.

Page 16, bottom; Page 17 top: The inclusion of variation for biodosimetry, especially for specific methods, is irrelevant (and misleading) – the simulations and the kriging are based solely on physical dosimetry.

Page 19; para 3: This discussion is useful but misses the point. If one can only practically process 160 individuals by biodosimetry (the time frame does not permit going back for a second sample), then one must focus on those most likely to be shown to irradiated (> 2 Gy). And one must have this subset within a few days at most (allowing how many rounds of physical dosimetry for kriging?). Beyond that, one is too late to initiate cytokine therapy with any expectation of success. Also unaddressed is whether the proposed biodosimetry techniques are effective for assessing cumulative chronic doses (over a few weeks) – or whether they have all been developed using a specific (24 hr?) time point of sampling after a single acute radiation exposure. If that is the case, then even 3-4 days for physical dosimetry kriging may not be beneficial

A final major issue that is subject to misinterpretation by an audience that is not familiar with the radiation countermeasures/response field is contained in the last sentence of Figure legend 1. ”The individuals residing in this part are eligible to be treated by cytokine therapies.” This implies that treatment (with potentially deleterious effects) would be administered to individuals on the basis of either a physical dosimetry measurement map that includes their location or even a biodosimetry assessment of a neighbor – but not themselves. This is most likely to be the case. Or if the authors, believe that it would be, they should provide a citation for the strategy.

In all, the authors present what could be a valuable aide in directing triage and treatment of selected individuals in the aftermath of a nuclear scenario. But it must be presented reasonably and without the potential to confuse and mislead a readership that will be predominately be unfamiliar with any radiation dosimetry methods or with the strategies likely to be employed in such a situation.

Reviewer #2: The authors use geospatial modeling in an attempt to decrease the need for radiation dosimetry in the setting of a large scale disaster. They modeled 30 scenarios, including 22 urban/high-density and 2 rural/low-density scenarios under various weather conditions. Multiple (3-10) rounds of sampling and kriging were required for the dosimetry maps to converge, requiring between 73 and 417 samples for different scenarios. On average, 70±10% of locations where populations are expected to receive an exposure ≥2Gy were identified.

They conclude that geostatistical mapping limits the number of individuals requiring dose assessment, the time required, and radiation exposure to first responders. Geostatistical analysis will expedite triaging of acute radiation exposure in population-scale nuclear events.

This manuscript is a highly mathematical approach to a major problem in a mass casualty event concerning radiation exposure. The major hurdle with this reviewer is that this approach does not really solve the problem. Based on their data from table 1, the range is so variable that this reviewer does not see the value of this approach. For example, the accuracy is as low as in the 20% range making this approach inferior to a coin toss. The best case scenario is in the low 80% range. I am not clear that this provides much more information since it’s not clear what the responders would approach the other 20% that did receive a higher dose.

Lastly, a major difficulty is partial body shielding which would make this approach even more difficult to interprete.

7. PLOS authors have the option to publish the peer review history of their article (what does this mean?). If published, this will include your full peer review and any attached files.

Reviewer #1: No

Reviewer #2: No

---

## [Author Response · Author response to Decision Letter 2]

26 Feb 2020

Summary of New Data Analyses added to Version 3 of the manuscript (for Editor and Reviewers):

1. We computed the accuracy of the derived plumes at the 3Gy radiation level. We hypothesized that because regions more distant from the epicentre of the nuclear event were less densely populated and less compact at the 2 Gy contour compared to the 3 Gy contour in some scenarios that the performance of the proposed method might improve at the higher radiation threshold. Plumes at the 3Gy level were similar to the HPAC plume in many scenarios (see Table 1, rightmost column), and this is described in the Results.

2. The initial conditions for stopping our procedure (>90% concordance between areas of consecutive plumes) were specified for rapid computation of an initial radiation plume map in a triage situation. We determined whether these convergence criteria were optimal using a higher stringency of overlap between the plumes of consecutive iterations of geostatistical analysis (>99% concordance) for one replicate in each scenario. The accuracy for some scenarios improved significantly (supplementary table S4, Figures S2 and S3), and this is summarized in the Results.

3. We estimated the magnitude of 2 Gy plume error by varying the contour threshold of the derived plume until the differences between the HPAC and derived plumes were minimized. Based on Reviewer #2 comments, a significant amount of the error reported for each scenario replicate is relatively close to the desired threshold. For example, 63% of Boston MA replicate #3 is accurate at the 2Gy contour, but 86% of the missing plume area occurs within 1.7-2.0 Gy exposure level. This is now described in the Discussion.

Comments to the Author

1. If the authors have adequately addressed your comments raised in a previous round of review and you feel that this manuscript is now acceptable for publication, you may indicate that here to bypass the “Comments to the Author” section, enter your conflict of interest statement in the “Confidential to Editor” section, and submit your "Accept" recommendation.

Reviewer #1: (No Response)

Reviewer #2: All comments have been addressed

2. Is the manuscript technically sound, and do the data support the conclusions?

Reviewer #1: Partly

Reviewer #2: No

 Response:

We note that Reviewer #2 has changed their decision from Yes in the previous two revisions of the manuscript. Based on Reviewer #2 comments in this most recent review, we assume this is due to the accuracy values presented in the manuscript, which have now changed since the initial draft of the manuscript. The particular replicates that the Reviewer has commented on consist of small numbers of samples in two scenarios with low density populations predominantly in rural areas. We have addressed this by performing additional simulations with increased initial population densities (0.2, 1.0%) and adding additional iterations of kriging and densification. Both of these have improved the final accuracies for these replicates. 

3. Has the statistical analysis been performed appropriately and rigorously?

Reviewer #1: Yes

Reviewer #2: Yes

4. Have the authors made all data underlying the findings in their manuscript fully available?

Reviewer #1: Yes

Reviewer #2: Yes

5. Is the manuscript presented in an intelligible fashion and written in standard English?

Reviewer #1: Yes

Reviewer #2: No

Response:

Documentation of specific errors as requested by PLOS ONE has not been provided by Reviewer #2. 

Based on the evidence after multiple rounds of review, reviewer #2 clearly understands the main points addressed in the manuscript, that is, the paper is written in an intelligible fashion. After multiple revisions, this Reviewer has not listed any errors in our grammar, spelling or clarity of presentation. 

6. Review Comments to the Author

Reviewer #1: 

1. This manuscript is definitely getting better; however, there remain problematic issues that relate to the potential for a causal reader to be misled as to what was done and what the results mean.

Response:

On multiple occasions, this Reviewer has indicated that we have mislead or have the potential to mislead the casual reader regarding terminology related to the types and methods of processing samples. We do not share this view. Nevertheless, have modified our manuscript to be agnostic about the type of dosimetry that can be used with the method we propose. We sincerely believe that this latest revision alleviates any remaining issues that led to their perception. We discuss this in greater detail in several of the responses below.

2. The Abstract is now an appropriate description of what was done. A minor suggestion is to move the first sentence of the Methods to before the current third sentence (Initially …). This would emphasize that this is a modelling exercise and that exposures or boundaries were not actually determined.

Response:

We agree to your suggestion and have changed the Abstract accordingly.

“Methods: Physical radiation plumes modelled nuclear detonation scenarios of simulated exposures at 22 US locations. Models assumed only location of the epicenter and historical, prevailing wind directions/speeds. The spatial boundaries of graduated radiation exposures were determined by targeted, multistep geostatistical analysis of small population samples...”

3A. One remaining issue in the manuscript body is in the overlap or inappropriate implied equivalence of “individuals” and “locations”. The authors should confirm (by searching for the words) that when talking about measurements or sampling or results that location is applied to physical dosimetry and individual is restricted to biodosimetry. Some of these may just be residual from prior versions but need to be corrected. This would then allow the appropriate inference by the reader that locations are fixed and may accurately reflect physical dose levels at or in proximity to that location. This then leads to a reasonable assumption that interpolation between two locations (by additional sampling) is likely to refine dose boundaries. 

Response:

The changes made to the manuscript in our previous revision were to generalize the text so that it could apply to any type of dosimetry method, as long as all measurements in a dosimetry map were obtained using the same method. We state that we are simulating “physical or cytogenetic dosimetry”, as dosimetry data from either method would result in an accurate plume. We have gone through the manuscript again in an attempt to further generalize the text to avoid any confusion about our presentation. Most uses of the term “individuals” have been changed to “samples.” In the Methods section, we define the term “samples” to refer either emissions measured by physical radiation detectors at the time of the nuclear incident or to radiation levels absorbed by biological samples. Since there could be differences between the radiation levels obtained by different methods, we indicate that the dosimetry maps are assumed to be consistently derived by the same method.

 The proposed geostatistical method can only be used with dosimetry at known physical locations. This is feasible by design of physical detectors containing GIS devices. In the case of biological samples, the random access memory of cellular telephones carried by individuals should still retain of the last known locations, even in the event of an electromagnetic storm that affects network function. Only biological samples remaining at a fixed location after the nuclear incident would be eligible for geostatistical analysis. The advantages and drawbacks of different dosimetry methods (see next section) do not negate the core claim of this manuscript that the geostatistical method proposed would reduce the required sampling to develop a sufficiently accurate plume for triage purposes. 

Table 1 indicates the number of unique samples used to derive our plume. We would like to point out that the number of samples in this table has been decreased significantly. This was due to identical samples being counted as unique if separate densification steps had selected the same location more than once. This was corrected, which reduced the number of samples (from a range of “73 - 417” to “58 – 347”).

3B. The same cannot be said for individuals. A value for a third individual, sampled at a location between the locations where two initial individuals were assessed by biodosimetry, would not a priori be expected to provide an intermediate biodosimetry value. This could be due to heterogeneity in inherent radiosensitivity of the population, movement by the individual, comorbidities, or any of the reasons that the authors allude to. This also leads into the need to be precise as to what measurements are reasonable to utilize for the kriging process and the re-iterative process. That is, one can reasonably postulate additional physical dosimetry sampling over a few days to refine dose boundaries that then can be used to perform biodosimetry on a selected set of individuals. But with the time restraints of biodosimetry it is not practical to consider more than one sampling as contributing to the triaging process. But current wording might suggest otherwise. As an example, at the bottom of page 3, there is the following:

“One approach to alleviating the need to triage all potentially exposed individuals would be to survey a subset of individuals combined with their respective locations by geostatistical analysis. We demonstrate that combining such surveys with the geolocations of these measurements can reduce sampling requirements in population-scale radiation scenarios and would be expected to decrease total radiation exposures of first responders.”

The term “combining” might be interpreted as using the two dosimetry methods reiteratively. Also “surveys” of individuals might be mis-interpreted as multiple surveys (i.e., biodosimetry) in the same general vicinity to aid in the kriging, rather than a single targeted survey of individuals in a location selected by the physical dosimetry kriging.

Response:

Each dosimetry method has advantages and drawbacks but determining the best type of dosimetry method to be used is not (and has never been) the purpose of this study. The objective of the paper is to find a way to accurately map radiation with limited sampling. Our approach could accelerate treatment decisions in a way that has not been previously considered. 

There are no optimal dosimetry methods. Applications of biodosimetry may be limited due to the total time required to obtain results, but it does not diminish the value of those results. Complete reliance on physical dosimetry may be concerning, since it does not measure the amount of radiation that has been absorbed and may also be prone to false positives. From the revised Discussion:

“Biodosimetry estimates absorbed exposures, whereas physical dosimeters measure environmental emissions. Physical dosimetry is more rapid and can map changing radiation plume locations dynamically. However, unfiltered radiation emissions are prone to false positive readouts, for example in aerial physical dosimetry counterterrorism surveys (Karam, 2017) due to common environmental sources of radiation. Uncorrected, such data will introduce errors and distort geostatistical-derived plumes. Mitigation may be possible by specifying the locations of radiation detectors by iterative kriging and densification. Nevertheless, biodosimetry at specified locations may provide results that might be useful in assessing treatment eligibility, especially in instances of borderline clinical exposures.”

We specify in the Methods section that sampling of consecutive iterations should use the same approach at each stage of refinement to derive an accurate plume. Our intent is to generalize the text in the manuscript to be agnostic to the dosimetry method used to derive a plume. Changes to the paragraph rectify this as well.

“One approach to alleviating the need to triage all potentially exposed samples would be to survey a subset of samples and their respective locations by geostatistical analysis. This survey may involve location-based physical dosimetry, where high-risk individuals are tagged based on their proximity to nearby detectors. We demonstrate that combining surveys using uniform methodologies with the geolocations of these measurements can reduce sampling requirements in population-scale radiation scenarios. This would also be anticipated to decrease overall radiation exposures to first responders.”

4. Last sentence of Introduction: “kriging using a relatively small number of samples analyzed by physical or biodosimetry methods.” Biodosimetry needs to be removed, but the sentence can be extended to state using kriging of physical dosimetry to aid in selection of individuals for biodosimetry.

Response:

As previously mentioned, we have decided to simplify the text so that the text does not specify the dosimetry method used, as a plume can be derived with any type of dosimetry method as long as it is consistent. We have therefore changed the last sentence of the Introduction to the following:

“The question this paper addresses is whether the radiation plumes derived by HPAC can be reconstructed with iterative kriging using a relatively small number of samples consistently analyzed by the same dosimetry method.”

5. Page 5: “Densification is the geostatistical procedure that targets and localizes an additional small cohort of irradiated individuals (1) to mitigate uncertainty in environmental measurement. These kriging and densification processes are repeated for a limited number of iterations until the coverage area and the radiation level contours of the inferred plume stabilizes (i.e. sampling of additional individuals (2)”. (1) it identifies a location, not any individuals who might be there. (2) Locations will be sampled reiteratively, not individuals.

Response:

. The text has been generalized in the following manner: 

- For issue (1): We have changed “irradiated individuals” to “sampling locations”. 

- For issue (2): We change “sampling of additional individuals” to “additional sampling”.

This text is now:

“Densification is the geostatistical procedure that targets and localizes an additional small cohort of sampling locations to mitigate uncertainty in environmental measurements. These kriging and densification processes are repeated for a limited number of iterations until the coverage area and the radiation level contours of the inferred plume stabilizes (i.e. additional sampling in the affected area does not significantly alter the geographic coverage of the plume or the estimates of absorbed radiation dose).”

6. Page 7, para 2: “To map predicted radiation levels based on the distribution of dose-estimated patients around the prediction location” It is locations sampled by physical dosimetry that have a distribution – not patients.

Response:

To generalize this sentence, we have changed “patients” to “samples” as this includes physical dose measurements. The sentence now reads as follows:

“To map predicted radiation levels based on the distribution of samples across the location of interest, …”

7. Figure legend 2; last sentence “% of the individuals eligible for cytokine treatment” – locations (weighted for population density) where treatments of individuals might be necessary.

Response: 

As suggested, we have edited the sentence to state that we are computing a dose estimate for a location (weighted by population density), rather than for individuals specifically:

“… the converged plumes localized 80.3% and 75% of the locations (weighted for population density) for treatment-eligible radiation exposures in these scenarios. “

8. Page 15; para 2; sentence 1: “subset of irradiated individuals”. Subset of locations - there is no a priori assumption about irradiation of individuals. Sentence 2: unirradiated individuals. Sentence 3: irradiated individuals.

Response:

As previously mentioned, this does not preclude the proposed method from being used to derive radiation plumes based on biologically-determined dose, regardless of assumptions about processing time (see our response to point 11 below). As a compromise, rather than using the word “locations”, we have used the word “samples”, or have modified the sentence to remove the segment which used the word entirely.

9. Page 16; para 2; sentence 1: “incorrect sampling of potentially exposed individuals”. Page 16; para 2; last sentence: “fewer irradiated individuals”.

Response:

Both suggestions pertain to a paragraph discussing dosimetry measurement error. We have modified the first sentence of this paragraph to eliminate the text referring to exposed individuals:

“…it might be expected to lead to improper sampling.”

For the final sentence of the paragraph, we eliminated the word “individuals” and replaced it with “samples”.

10. Page 16, bottom; Page 17 top: The inclusion of variation for biodosimetry, especially for specific methods, is irrelevant (and misleading) – the simulations and the kriging are based solely on physical dosimetry.

Response:

In response to this reviewer’s previous comments, we revised the text to be agnostic to the type of dosimetry method used to obtain the data. As we have indicated (and cited), physical dosimetry is also prone to systematic sources of radiation measurement error. This justifies inclusion of the examples given in the text. We do not agree with the reviewer’s claim that inclusion of variation in radiation measurements is irrelevant and misleading. Indeed, the examples provided illustrate the robustness of the proposed methods to different sources of systematic error. The section on inferred radiation exposures under sub-optimal sampling conditions remains relevant, and we argue, is a strength of our approach. We have therefore included “physical dose estimation error” as an example of possible sources for variation in radiation dose measurements.

“Maximum deviations were designed to represent confidence values in physical and/or biodosimetry methods, including physical dose estimation error (29), dicentric analysis…”

11. Page 19; para 3: This discussion is useful but misses the point. If one can only practically process 160 individuals by biodosimetry (the time frame does not permit going back for a second sample),

Response:

The reviewer appears to assume that First Responses to a nuclear incident regarding sample acquisition and laboratory processing samples will be limited in scope and resources. However, in our previous response to this reviewer, we suggested that field, laboratory and computational resources could be amplified through parallel deployment of multiple dedicated teams and automation. Indeed, we cited a recent, peer-reviewed Concept of Operations article in Radiation Protection Dosimetry that recommended large scale processing and multiple procurement mechanisms. We suggest that the reviewer consider such alternatives, which could overcome the capacity and time limitations that s/he envisions in a large scale nuclear incident. 

11. (continued) then one must focus on those most likely to be shown to irradiated (> 2 Gy). And one must have this subset within a few days at most (allowing how many rounds of physical dosimetry for kriging?). Beyond that, one is too late to initiate cytokine therapy with any expectation of success. 

Response:

Our paper does not model how long dosimetry testing and measurement will require or what type of testing (physical or biological) will be performed. The reviewer’s assumptions about what would be feasible to accomplish within a 3 day time frame seem to be predicated on a single field team sampling exposed individuals or placing radiation detectors followed by sequential geostatistical analysis. With high performance computing and parallel processing, it should be possible to model different data sources at the same time, and then combine these into more robust geostatistical models. It is really beyond the scope of this first publication on the subject (nor do we have the resources to perform at this time) large scale modeling of multi-source data processing and geostatistical analysis. Nevertheless, we stand by the main conclusions of the paper, that strategic sampling guided by geostatistical methods of a relatively small fraction of irradiated “samples” can be used to compute exposures for much larger populations. 

11. (continued) Also unaddressed is whether the proposed biodosimetry techniques are effective for assessing cumulative chronic doses (over a few weeks) – or whether they have all been developed using a specific (24 hr?) time point of sampling after a single acute radiation exposure. If that is the case, then even 3-4 days for physical dosimetry kriging may not be beneficial

Response:

We state that biodosimetry does not address dynamic changes in radiation levels which have cumulative effects. Physical dosimetry can provide dynamic measurements of emitted radiation, but does not measure how much radiation is absorbed. In any case, distinctions about intermediate or long term exposures are not relevant in a triage situation where the existing testing capacity may not be sufficient to process every individual or sample that would be desirable to create a dense dosimetry plume map. 

12. A final major issue that is subject to misinterpretation by an audience that is not familiar with the radiation countermeasures/response field is contained in the last sentence of Figure legend 1. ”The individuals residing in this part are eligible to be treated by cytokine therapies.” This implies that treatment (with potentially deleterious effects) would be administered to individuals on the basis of either a physical dosimetry measurement map that includes their location or even a biodosimetry assessment of a neighbor – but not themselves. This is most likely to be the case. Or if the authors, believe that it would be, they should provide a citation for the strategy.

Response:

We concur that the derived map would not be used – by itself - as the sole basis of managing treatment. The described method is designed to triage and identify individuals who may benefit from treatment in situations, based on limited sampling of irradiated individuals in the affected geographic regions. The Introduction of the paper describes multiple clinical criteria that contribute to the diagnosis of Acute Radiation Syndrome, and nothing in the paper disputes any of these criteria. Besides symptoms, laboratory testing would contribute valuable confirmation of these symptoms (especially if they were mild or non-specific due to presence of other confounding diagnoses) may treatment, which may include cytokine therapies.

To avoid any confusion or misinterpretation, we have decided to delete this sentence entirely from the Figure 1 legend, as it is unrelated to the content of Figure 1. Furthermore, the significance of the >2Gy threshold has been stated elsewhere in the manuscript.

13. In all, the authors present what could be a valuable aide in directing triage and treatment of selected individuals in the aftermath of a nuclear scenario. But it must be presented reasonably and without the potential to confuse and mislead a readership that will be predominately be unfamiliar with any radiation dosimetry methods or with the strategies likely to be employed in such a situation.

Response

We modified the paper according to the reviewer’s recommendations to avoid endorsement of any particular dosimetry method. We believe that those deciding what strategy to employ after a nuclear incident should not be bound by previous dogma about which types of dosimetry to perform and when. Further research in this area could address optimization of geostatistical analysis when multiple dosimetry data types are collected. 

However, the purpose of this study was to introduce this approach and demonstrate using simulations that it could provide a means of inferring radiation across the landscape given a limited set of assumptions (epicentre, wind vector, and weather), and fewer radiation measurements. For us to provide a complete TR7-level solution at this stage would be beyond the scope of this paper. 

Reviewer #2: 

The authors use geospatial modeling in an attempt to decrease the need for radiation dosimetry in the setting of a large scale disaster. They modeled 30 scenarios, including 22 urban/high-density and 2 rural/low-density scenarios under various weather conditions. Multiple (3-10) rounds of sampling and kriging were required for the dosimetry maps to converge, requiring between 73 and 417 samples for different scenarios. On average, 70±10% of locations where populations are expected to receive an exposure ≥2Gy were identified. They conclude that geostatistical mapping limits the number of individuals requiring dose assessment, the time required, and radiation exposure to first responders. Geostatistical analysis will expedite triaging of acute radiation exposure in population-scale nuclear events.

This manuscript is a highly mathematical approach to a major problem in a mass casualty event concerning radiation exposure. The major hurdle with this reviewer is that this approach does not really solve the problem. Based on their data from table 1, the range is so variable that this reviewer does not see the value of this approach. For example, the accuracy is as low as in the 20% range making this approach inferior to a coin toss. The best case scenario is in the low 80% range. I am not clear that this provides much more information since it’s not clear what the responders would approach the other 20% that did receive a higher dose.

Response:

We reject the notion our “mathematical approach“ should be expected to completely identify every single individual with clinically relevant radiation exposure levels. We anticipate that it will be primarily applied in triage situations soon after a nuclear incident, with the primary focus of providing guidance for an orderly and efficient response in the early stages of in a large scale nuclear incident. 

The reviewer misstated the performance of our best case scenario which was actually 90.7% overlap of the Burlington VT HPAC plume at the 2Gy plume contour. The current version of the manuscript now demonstrates that the accuracy of overlap between plumes –where it counts- at higher radiation levels (3 Gy) is higher for most of the scenarios that we present. Nearly all of the contours are contiguous and non-intersecting, that is 3 Gy exposures which are miscalled are still interpreted as >2 Gy. Although incorrectly determined high level exposures are slightly underestimated, such individuals will still be brought to clinical attention. 

To the best of our knowledge, this is the first report of this approach for triaging radiation exposed individuals for sample measurement. While it may not be perfect, we present a novel, useful and scientifically valid solution to a longstanding problem. Perfection is likely unachievable. Indeed, as we describe, physical radiation measurements and biodosimetry testing also contribute systematic error, and therefore even the most comprehensive saturation testing protocols will be susceptible to such errors. 

We have revised and ensured that this paper meets the criteria for publication in PLoS ONE. Specifically, the simulations we performed have been conducted rigorously, with appropriate replication, and we have presented a wide variety of scenarios to explore the strengths and weaknesses of the proposed approach. The data presented in the manuscript support the conclusions drawn and are not overstated. Methods and data have been deposited in publicly available resources. In fairness, we ask the Reviewer to consider the publication criteria that the journal has established in assessing our contribution. 

We have been honest and transparent about sources and impacts of error in radiation measurements and limitations of the approach for computing accurate low level radiation exposures. For example, the census data that is publicly available is not as granular as the data that the US Census Bureau actually collects. Thus, when we compute the number of samples that would be impacted by a radiation plume, this requires us to compute the population density for the minimal reported geographic region, which is usually a county subdivision. This averages impact over a geographic area and affects the accuracy of our estimates. Small overlapping segments of subdivisions are particularly prone to these types of errors. Also, we show that the reported accuracy does depend on the fraction of the total population that is initially sampled, and that sparsely populated regions are more prone to lower levels of accuracy (see below). 

While the majority of scenarios gave results consistent with the HPAC plume, there were 2 exceptions. We have performed additional work that provides an explanation of the particular Columbia SC and Columbus OH anomalous replicates which did not reproduce the HPAC plume (Results, 3rd paragraph, Discussion 2nd paragraph). Initial sampling for these replicates exhibited sparse coverage over large regions of the HPAC plume. Densification steps did not select locations for further sampling. Scenarios with low population density appear to be susceptible to this problem. We demonstrate that it can be mitigated by either increasing sampling density, or by strategically selecting sampling locations within this region. 

From the Discussion:

“In a real-world scenario, secondary sampling locations assigned by densification would be supervised, which would direct the software towards derivation of a complete and accurate plume. Indeed, we found that manually adding two new sample locations to the unrepresented region corrected the poor results obtained for the Columbia SC scenario replicate after two additional cycles of kriging and densification. “

Lastly, a major difficulty is partial body shielding which would make this approach even more difficult to interpreted.

Response:

Reviewer #2 had previously mentioned shielding during the first round of revisions:

“The major concern is that the radiation doses in a large city will be inhomogeneous due to partial shielding. How does this algorithm account for such events?”

To which we responded:

The Discussion describes the limitations of our method. Here, we discuss how shielding (or “shadowing”) by infrastructure is not accounted for, and how these concessions may limit the overall accuracy of our method. 

Results (Paragraph 3, page 18): “Finally, neither the HPAC version available for this study, nor our geostatistical models account for shielding by infrastructure, such as shadowing, which computes the degree to which radiation is prevented from reaching certain locations by the urban environment (35). While these factors will impact the predicted accuracies of derived dosimetry maps, these effects will also have to be accounted for in ground truth models, such as HPAC, before they could be addressed in geostatistical interpolation.”

During the second round of revisions, it seemed that this response had satisfied the reviewer:

“Reviewer #2: all concerns addressed - no further comments”

In the future, obstacles that occlude diffusion of radioactive particles could be factored into the derived plume. Samples from city neighborhoods with densely arrayed, tall building infrastructure may be partially shielded, resulting in lower actual exposures to ionizing radiation. However, we did not and still do not have access to these calculations to test this possibility, since these algorithms and software, to the best of our knowledge, are not publicly available.

---

## [Decision Letter · Decision Letter 3]

12 Mar 2020

PONE-D-19-22325R3

Meeting radiation dosimetry capacity requirements of population-scale exposures by geostatistical sampling

PLOS ONE

Dear Dr, Rogan:

Thank you for submitting your manuscript to PLOS ONE. After careful consideration, we feel that it has merit but does not fully meet PLOS ONE’s publication criteria as it currently stands. Therefore, we invite you to submit a revised version of the manuscript that addresses the points raised during the review process.

Additional minor comments have been suggested by one reviewer.  Please address these in a revision.

We would appreciate receiving your revised manuscript by Apr 26 2020 11:59PM. To enhance the reproducibility of your results, we recommend that if applicable you deposit your laboratory protocols in protocols.io, where a protocol can be assigned its own identifier (DOI) such that it can be cited independently in the future. For instructions see: http://journals.plos.org/plosone/s/submission-guidelines#loc-laboratory-protocols

We look forward to receiving your revised manuscript.

Kind regards,

Gayle E. Woloschak, PhD

Academic Editor

PLOS ONE

Additional Editor Comments (if provided):

One reviewer has suggested some minor edits for the work in the comments below.

Reviewers' comments:

Reviewer's Responses to Questions

**Comments to the Author**

1. If the authors have adequately addressed your comments raised in a previous round of review and you feel that this manuscript is now acceptable for publication, you may indicate that here to bypass the “Comments to the Author” section, enter your conflict of interest statement in the “Confidential to Editor” section, and submit your "Accept" recommendation.

Reviewer #1: (No Response)

Reviewer #2: All comments have been addressed

2. Is the manuscript technically sound, and do the data support the conclusions?

Reviewer #1: Partly

Reviewer #2: Yes

3. Has the statistical analysis been performed appropriately and rigorously? 

Reviewer #1: N/A

Reviewer #2: Yes

4. Have the authors made all data underlying the findings in their manuscript fully available?

Reviewer #1: Yes

Reviewer #2: Yes

5. Is the manuscript presented in an intelligible fashion and written in standard English?

Reviewer #1: Yes

Reviewer #2: Yes

6. Review Comments to the Author

Reviewer #1: Still a few instances where the term “individuals” is inaccurate or misleading. These are easily corrected without changing the meaning of the sentence.

Page 2; para 4; line 6-7: “limits the number of individuals requiring dose assessment”. Suggest re-wording as “limits the number of required dose assessments”

Page 4; para 3; line 2-3: “subset of radiation exposed individuals or locations”.

Page 6; para 3; line 8: “locations of either radiation-exposed individuals or physical radiation detectors”. The problem here is not specifically with individuals but with the conclusion that they are all radiation-exposed (which assumes knowledge prior to testing). Moreover, it is acknowledged that most physical dosimetry readings will be zero. For simplicity, can the phrases be replaced with “subset of individuals or locations” and “locations for dose assessment”, respectively? The phrase is also used on page 18 (last line) but is not so critical there.

Page 21; para 2; line 8-9: “the number of tested individuals necessary for derivation of an accurate plume” Replace individuals with samples.

Lastly, there is an apparent discrepancy which even if the statements are correct could confuse the reader. On page 6; para 3; line 6 it is stated that “random samples, which corresponded to 0.1% of the population of each sub-division”. On the next page (para 2; line 6) there is a similar statement “random points representing 1% of the population of the … subdivisions”. Should it be 0.1% in both places? That would make the value of 223/617 more reasonable than 223/6175. If the text is correct, a few additional words of explanation might be helpful. 0.1% is also used on page 18

Because there have been modifications to the data and how they are presented, I re-examined the tables (table 1 plus supplementary) for clarity and consistency.

Page 6; para 2; line 11-12: “(topological contours range from 1.0-7.0Gy in intervals of 0.5Gy).” From Figure 2 it appears that the intervals are 1 Gy.

Page 7; para 1; line 5-6: “The number of random samples generated for each scenario, and how many of those overlapped the HPAC plume, is available in S1A Table”. It would be helpful to place here the statement in the next paragraph that “overlapped” equates to samples >0Gy.

Page 10; para 1; line 1 (Figure 2 legend): “The total number of samples in one iteration is indicated (in parentheses)”. It seems that the numbers shown are not total samples but rather samples >0 Gy.

Page 11; para 2; line 7: The potential for confusion mentioned in the above item is repeated here. The phrase used of “irradiated samples” is imprecise. No samples were irradiated; rather, what is meant here is samples with a dose greater than 0 Gy. The potential for confusion is enhanced when in Table 1 the column header is simply “No. of Samples” without reference to these being greater than 0 Gy. The same appears in Table S2.

Page 15; para 3; line 3: “majority of these locations did not overlap with the HPAC plume and have therefore been modelled as unirradiated samples.” Majority is an understatement. From table S1A, the lowest frequency of values generated as 0 Gy was 94.5%. But in some regions (Chicago) it can be 99%. Which would seem to suggest that in these scenarios the first iteration might provide the outer boundaries (for any dose) but would provide little in the way of interior contour structure. The situation is even worse when one starts with a low number in addition to low frequency. For Cincinnati, with 2 points with dose (0.5%), how can any contour be derived? The result seems to be that a larger number of iterations is then required to derive the final plume. But then this doesn’t hold true, with 6 and 8 iterations required when starting with two points, yet 9 iterations when starting with 9 dose points. On the other hand, Chicago consistently required only 4 iterations (and a low number of samples) to reach the final plume [Or is this something trivial such as all the wind blows east and it is easy to model doses in the lake where there is no population?]. A critical aspect – in a disaster scenario – would be the number of iterations (and the time) required to arrive at a final plume. Is there any analysis that could assist in identifying factors (other than initial sampling size) that could be used to reduce the number of iterations?

Considering the two comments above, it would seem that a useful set of data would be the actual numbers of both samples with dose and samples without dose that are added at each iteration step. In other words, how does one go from 2 and 393 (first step for replicate 1 for Cincinnati) to 139 and ??what is the total number of samples with zero dose that end of being selected?? One would expect that with each iteration there are proportionally more samples with dose and fewer without dose. But the “improvement” at each step, and the total number of samples that have to be assessed in a real-word scenario are critical aspects of the potential utility of this approach. As it is, the reader is unable to evaluate the overall requirements. I might argue that this information is more meaningful that the replicates in Table 1 (which could be presented in a supplemental table).

Figure 2A: The numbers shown for Samples at first and final iteration (assuming >0Gy), BCD and RMSD for final vs HPAC plume do not agree with values provided in Table 1 and S1A. For B (Albany), only the sample number in first iteration (12) agrees with table S1A and Table 1.

In summary, this paper has the most value if it is clear to the naïve reader what was done and how the strategy might be used to advantage. And for a reader knowledgeable in the field who wants to examine the process in detail, there must be sufficient accurate and consistent data to allow that.

Reviewer #2: no further comments - all addressed

xxxxxxxxxxxxxxxxxxxxxxxxxxxxxxxxxxxxxxxxxxxxxxxxxxxxxxxxxxxxxxxxxx

7. PLOS authors have the option to publish the peer review history of their article (what does this mean?). If published, this will include your full peer review and any attached files.

Reviewer #1: No

Reviewer #2: No

---

## [Author Response · Author response to Decision Letter 3]

29 Mar 2020

Reviewer #1: Still a few instances where the term “individuals” is inaccurate or misleading. These are easily corrected without changing the meaning of the sentence.

Page 2; para 4; line 6-7: “limits the number of individuals requiring dose assessment”. Suggest re-wording as “limits the number of required dose assessments”

RESPONSE

The change has been incorporated.

Page 4; para 3; line 2-3: “subset of radiation exposed individuals or locations”.

Page 6; para 3; line 8: “locations of either radiation-exposed individuals or physical radiation detectors”. The problem here is not specifically with individuals but with the conclusion that they are all radiation-exposed (which assumes knowledge prior to testing). Moreover, it is acknowledged that most physical dosimetry readings will be zero. For simplicity, can the phrases be replaced with “subset of individuals or locations” and “locations for dose assessment”, respectively? 

RESPONSE

These changes have been incorporated to both locations of the manuscript.

The phrase is also used on page 18 (last line) but is not so critical there.

RESPONSE

The last line of this paragraph has been changed to the following:

“After a nuclear incident, processing all individuals for dose assessment has been acknowledged to be labor intensive, and would likely be a major bottleneck in identifying those who require immediate treatment (32). “ 

Page 21; para 2; line 8-9: “the number of tested individuals necessary for derivation of an accurate plume” Replace individuals with samples.

RESPONSE

The suggested change has been incorporated.

Lastly, there is an apparent discrepancy which even if the statements are correct could confuse the reader. On page 6; para 3; line 6 it is stated that “random samples, which corresponded to 0.1% of the population of each sub-division”. On the next page (para 2; line 6) there is a similar statement “random points representing 1% of the population of the … subdivisions”. Should it be 0.1% in both places? 

RESPONSE

No. The 1% sampling rate on page 7 was only used to compare the results of the different kriging approaches. Once the EBK was determined to be the best approach, initial sampling was performed at 0.1% of the population as indicated on page 6. We have clarified the text on page 7 to make the reasons for sampling at 1% clearer to the reader. 

That would make the value of 223/617 more reasonable than 223/6175. If the text is correct, a few additional words of explanation might be helpful. 0.1% is also used on page 18

*RESPONSE

The default initial sampling rate for all scenarios was 0.1% of the population. However, Table 1 also shows the results at higher initial sampling rates tested for the Columbia SC (0.2% and 1.0%) and Columbus OH (0.2% only) scenarios, which significantly improved the performance of the method in these cases. As explained in the manuscript, these regions exhibited strong inhomogeneity in population densities, especially in regions of the plume that were more remotely located from the epicentre of the event. We believe this is clearly explained in our previous response to this reviewer and in the manuscript itself. 

Because there have been modifications to the data and how they are presented, I re-examined the tables (table 1 plus supplementary) for clarity and consistency.

Page 6; para 2; line 11-12: “(topological contours range from 1.0-7.0Gy in intervals of 0.5Gy).” From Figure 2 it appears that the intervals are 1 Gy.

*RESPONSE

In this paragraph, we are describing the raw HPAC data, which indeed consists of contours at intervals of 0.5Gy. For clarity, however, we set ArcMap to only display the contours at 1.0Gy, as displaying all contours (N=13 total) made the plumes difficult to interpret visually.

We want to make it clear that this decision has absolutely no impact on the overall accuracy of the derived plumes. When assigning our simulated sampling locations a radiation value based on its location relative to the HPAC plume, this included all contours (including 1.5Gy, 2.5Gy, etc).

Page 7; para 1; line 5-6: “The number of random samples generated for each scenario, and how many of those overlapped the HPAC plume, is available in S1A Table”. It would be helpful to place here the statement in the next paragraph that “overlapped” equates to samples >0Gy.

RESPONSE

The suggested change has been incorporated.

Page 10; para 1; line 1 (Figure 2 legend): “The total number of samples in one iteration is indicated (in parentheses)”. It seems that the numbers shown are not total samples but rather samples >0 Gy.

*RESPONSE

As requested, this is now indicated in the figure legend. 

Please note that this was already clear, as previous text had already addressed this issue:

p.7 (para 2) “of which 223 points overlapped the Boston HPAC plume (predicted dose >0Gy” and “A high number of unirradiated (0Gy) samples can depress the range of the plume; therefore, these locations were restricted to the subdivisions immediately surrounding the irradiated region.”

p.8 (para 1) “As a consequence, the process often did not always yield 200 unique samples with values exceeding 0Gy”

Page 11; para 2; line 7: The potential for confusion mentioned in the above item is repeated here. The phrase used of “irradiated samples” is imprecise. No samples were irradiated; rather, what is meant here is samples with a dose greater than 0 Gy. 

RESPONSE

The change has been incorporated.

The potential for confusion is enhanced when in Table 1 the column header is simply “No. of Samples” without reference to these being greater than 0 Gy. The same appears in Table S2.

*RESPONSE

Tables 1 and S2 are first referenced on page 11. The limitation of samples to those >0Gy is described on pages 7 and 8 (see response to above comment). Although the predicate statements are sufficiently clear regarding the results shown in these Tables, we have incorporated the change requested by the reviewer.

Page 15; para 3; line 3: “majority of these locations did not overlap with the HPAC plume and have therefore been modelled as unirradiated samples.” Majority is an understatement. From table S1A, the lowest frequency of values generated as 0 Gy was 94.5%. But in some regions (Chicago) it can be 99%. Which would seem to suggest that in these scenarios the first iteration might provide the outer boundaries (for any dose) but would provide little in the way of interior contour structure.

*RESPONSE

This lack of definition of the plume in the first iteration is evident from the Albany example in Figure 2 in the manuscript. It is important to recall that the only assumptions are the location of the epicentre and the approximate wind vector, so there is no expectation that it will be accurate at this stage. The subsequent steps identify the map locations with lowest confidence (highest variance) radiation estimates to define the locations for sampling in the next round of densification. See text below which address how these locations are chosen (page 8).

Furthermore, please note that our method uses ArcMap software to select initial random points evenly across a subdivision. In some scenarios, the plume only encompasses a small portion of the total sub-division area (e.g. the Boston MA scenario), which is why some scenarios have a high proportion of 0Gy samples. In a real-world scenario, sampling could be limited to a considerably smaller area based on preliminary information (e.g. wind direction). This is why when discussing testing unirradiated samples in the Methods, we mention that aerial survey could help target initial sampling regions: 

“We envision that testing could be greatly reduced by initially measuring background or low level physical radiation in population scale events by aerial surveys….”

p. 7 of Methods. This statement has been included in the past several revisions of the manuscript but has been updated to improve clarity.

 The situation is even worse when one starts with a low number in addition to low frequency. For Cincinnati, with 2 points with dose (0.5%), how can any contour be derived? 

*RESPONSE

This is actually one of the strengths of the geostatistical approach. So long as there are at least two samples with exposure >0 Gy, densification can derive the locations of additional samples in subsequent iterations. 

“There are instances in which 2 irradiated samples were adequate to progress plume development (e.g. Cincinnati urban sampling #2 [Table 1]).

We also demonstrate this by intentionally mis-specifying the wind vector in the section titled “Inferred Radiation Exposures Under Suboptimal Sampling Conditions” on page 17:

“The 0.05:0.2% ratio simulates a wind measurement error of 29.1º north (or N 22.8º W) relative to the actual wind direction of the HPAC plume. The 0.01:1.0% corresponds to a deviation of 40.9º north (or N 11.0º W). Despite this initial sampling error, inferred radiation plumes comparable to the correct plume were obtained.”

In a previous revision of this article, the reviewer requested that we eliminate this section of the manuscript. The reviewer seems to misunderstand this aspect of geostatistical dosimetry, but it clearly demonstrates that a plume can be defined even if only a small number of initial locations with dose estimates > 0 Gy have been identified within the derived plume. 

The result seems to be that a larger number of iterations is then required to derive the final plume.

RESPONSE

Usually this generalization is correct, but not in all instances, see below.

 But then this doesn’t hold true, with 6 and 8 iterations required when starting with two points, yet 9 iterations when starting with 9 dose points. On the other hand, Chicago consistently required only 4 iterations (and a low number of samples) to reach the final plume [Or is this something trivial such as all the wind blows east and it is easy to model doses in the lake where there is no population?].

RESPONSE

The distribution of samples is derived from the population density which is inhomogeneous in almost all scenarios (New York urban scenario, being an exception). Sampling in the proposed method is highly dependent on population density. Lake Michigan is never sampled because the US census doesn’t count any individuals in this location. This is a reasonable assumption whether physical- or bio-dosimetry is performed. 

The number of iterations required can vary in different replicates because different initial random locations are selected often in distinct county subdivisions with distinct population densities. Replicates have the same number of initial samples selected in the same subdivisions. However, where these samples overlap the HPAC plume, their locations can differ, which appears to have a significant impact on the final derived plumes.

A critical aspect – in a disaster scenario – would be the number of iterations (and the time) required to arrive at a final plume. Is there any analysis that could assist in identifying factors (other than initial sampling size) that could be used to reduce the number of iterations?

*RESPONSE

The number of iterations required depends on many factors, but most critically, the stopping criteria for convergence of the process. We used 90% overlap between consecutive plume iterations because the process described here was intended for triage purposes. Subsequent more stringent criteria (99% overlap) shown in the previous version of this manuscript required more iterations, but was also more comprehensive relative to the HPAC plume.

Sampling strategies would be more precise with more granular street-level US Census data. We would not be limited to selecting sampling locations based on average population densities across county subdivisions, which themselves, can be quite inhomogeneous. This is not possible with publicly available data sources. This would optimize the selection of sampling locations across the topographic map of the final plume. 

Considering the two comments above, it would seem that a useful set of data would be the actual numbers of both samples with dose and samples without dose that are added at each iteration step. 

RESPONSE

We are comfortable with using the 0Gy measurements obtained by the US Department of Energy Aerial Surveys that would be used in the geostatistical analysis that defines the outer limits of the radiation plume. There is an excess of such measurements in these models, in order to focus the statistical analysis on defining the plume in regions where radiation (absorbed or emitted) is evident. 

Our Zenodo archive now contains a file named “Progression-of-New-Densification-Selected-Sampling-Locations-For-All-Scenarios.xslx” which provides a categorical breakdown of how many unique densification-selected sampling locations occur within the irradiated region (i.e. overlap the HPAC plume) for each iteration of all scenario replicates. This Excel file is found within the archive “Intermediate-Derived-Plumes.Data-Points.zip”.

In other words, how does one go from 2 and 393 (first step for replicate 1 for Cincinnati) to 139 

*RESPONSE

This is addressed on page 8 (text bolded):

The “Densify Sampling Network” tool of the Geostatistical toolbox indicates lower confidence regions in the kriging-derived map, i.e. regions with highest variance specifying radiation dose [17]. We applied this tool to limit results to regions that would most likely exceed a pre-defined radiation level threshold. In practice, the locations selected by densification would be used to direct first responders to new locations for subsequent rounds of data acquisition in order to improve the accuracy of the kriging-derived map. Using 2Gy as the critical threshold (selection criterion QUARTILE_THRESHOLD_UPPER option), densification on one plume identified a maximum of 200 new sampling locations. Densification is a compute intensive step, requiring approximately 1 hour on a desktop with an Intel i7-4770 processor [3.4Ghz] and 16GB of RAM. Note that reducing the number of requested sampling locations decreases overall processing time. The Densify Sampling Network tool would sometimes select a sample at the same latitude and longitude between iterations. Furthermore, many densification-selected samples did not overlap the HPAC plume. As a consequence, the process often did not always yield 200 unique samples with values exceeding 0Gy. New sample data were assigned radiation values based upon their locations within the HPAC-generated plume, and kriging was performed on these and the original samples to generate another iteration of the inferred plume. 

We have previously addressed this particular issue during second revision of this manuscript. Our response to this reviewer’s comments e included a step-by-step breakdown of the derivation of a previously unreported replicate for the Columbus OH scenario. 

 “We find:

• Accuracy of plume from initial set of random samples only: 0% accuracy (the entire plume is <2Gy; image of plume in Section 4 of the online protocol) [N=8 samples > 0 Gy]

• Accuracy after 1 Iteration: 21.1% accuracy (image of plume in Section 5 of the online protocol) [N=55 samples > 0 Gy]

• Accuracy after 3 Iterations: 46.0% accuracy [N=79 samples > 0 Gy]

• Accuracy after 5 Iterations: 68.4% accuracy (final derived plume; image of plume in Section 6 of the online protocol) [N=136 samples > 0 Gy]”

and ??what is the total number of samples with zero dose that end of being selected??

*RESPONSE

Densification-directed sampling locations may indeed be located in regions outside of the HPAC plume. The fraction of densification-selected samples depends on many factors, including the stage of plume development, as densification selects points based on the quality of the plume it is given. However, selected sampling locations that are outside of the plume region could be ignored by our proposed method. Since the first revision of the manuscript, we included the following statement (which has been altered over the course of our revisions):

 (p.7, para. 2) “We envision that testing could be greatly reduced by initially measuring background or low level physical radiation in population scale events by aerial surveys or targeted multiplex dosimetry. “

For clarity, we now also state the following in the paragraph describing densification: 

(p.8, para 2) “We assume that locations within the 0 Gy envelope surrounding the plume do not have to be sampled in subsequent kriging iterations.”

One would expect that with each iteration there are proportionally more samples with dose and fewer without dose.

*RESPONSE

Anecdotally, we find that the highest ratio of irradiated/unirradiated samples are selected during the second densification (third iteration). The first densification step often includes locations in unirradiated regions (especially in low-density scenarios). The first densification is based on kriging following the initial sampling, and often produces a low definition plume. The plume from the following iteration is significantly improved because of the inclusion of more irradiated samples. We noticed that subsequent densification steps are prone to selecting the same locations as prior densification steps (as described in the Methods), meaning that the proportion of novel samples is reduced compared to earlier steps. Also, the locations of these novel samples tended to occur beyond the irradiated regions.

We have found that the fraction of irradiated to unirradiated sampling locations varies among scenarios and individual replicates for the same scenarios. Initial densification steps for scenarios in regions in high-density populations (e.g. urban New York and Washington D.C. scenarios) and were much more likely to select sampling locations in irradiated regions, while scenarios within regions of low population density (Charleston SC and Des Moines IA) had a greater proportion of locations selected in irradiated regions in later densification steps. 

We also find that this fraction can be influenced by differences between replicates for the same scenario. Replicates differ by consisting of an entirely different set of initial sampling locations (each location randomly selected by ArcMap software). For example, the first densification step for Albany NY scenario replicate #3 selected nearly 4 times as many sampling locations with dose compared to replicate #2 (N=67, 26 and 101 irradiated sampling locations selected by the first densification step for Albany NY replicate #1, 2 and 3).

Nevertheless, we are concerned about broad generalizations, because of the considerable variability in the locations and exposures of samples obtained through densification within different replicates of the same scenario and in different scenarios. Rather, the focus in this manuscript is to determine whether the scenarios converge to fulfill the criteria for discontinuation of the process and how many samples and iterations were required to meet these criteria.

 But the “improvement” at each step, and the total number of samples that have to be assessed in a real-word scenario are critical aspects of the potential utility of this approach. 

*RESPONSE

The total number of samples required at each step is shown in Figure 2. Superfluous sampling of unirradiated locations or individuals does not occur: the number of samples indicated in the figure is the actual number sampled. The incremental sampling reflects the difference between the numbers of samples at each successive iteration. For example, in panel B, 67 additional samples are evaluated in the 2nd iteration that were not included during the 1st iteration. The 3rd iteration requires 27 additional samples that were not available during the 1st and 2nd iterations. The level of effort required means that it should be feasible for first responders to procure these samples. With protective shielding and geospatial targeting, the limited amount of sampling would minimize their duration of their overall exposure to radiation.

As it is, the reader is unable to evaluate the overall requirements.

*RESPONSE

Based on the following, we disagree with the reviewer’s assessment that we have not provided sufficient details on our methods and results for the scenarios we have analyzed. The details of our procedure are provided – as requested by the editor during the second round of review – at Protocols.IO: (http://dx.doi.org/10.17504/protocols.io.ba4nigve; "Protocol for Geostatistical Determination of Radiation Dosimetry Maps of Population-Scale Exposures"). 

We have also included an extensive Zenodo archive of data and programs since the first revision of this manuscript (labeled for Peer Review Only, as it would only be published on Zenodo if the paper is determined to be accepted for publication). This information is cited in the above protocol and can be used to reconstruct our work including: 1) modified U.S. state and sub-division boundary files [in KML format], which can be imported into ArcMap using its KMLtoLayer function. These files have been modified to prevent sub-division naming issues that we encountered when importing boundary data into ArcMap; 2) contain HPAC plume coordinate (WGS1984) and dose (in cGy) values for all scenarios discussed in the manuscript. We provide "processed" and "unprocessed" HPAC plume data files. The "unprocessed" HPAC plume data are provided in its original XML format, which cannot be imported into ArcMap directly. The "processed" HPAC plumes are provided in tab-delimited X,Y,Z format (Latitude, Longitude, and Dose). We have also added a "0 cGy" contour in the "processed" plumes; 3) geostatistically-derived plume coordinate (WGS1984) and dose (in cGy) values for all scenarios discussed in the manuscript. Data is in comma-delimited format (Latitude, Longitude, and Dose).4) We have updated this component of the archive to include the sample locations and estimated doses for all iterations for each scenario as requested by the reviewer. Sample data consist of a set of coordinates generated at random locations within each Census sub-division using the ArcMap tool, ‘CreateRandomPoints_management’, and subsequent points generated by densification (the geostatistical procedure that targets and localizes an additional small cohort of irradiated individuals to mitigate uncertainty in environmental measurements.

 I might argue that this information is more meaningful that the replicates in Table 1 (which could be presented in a supplemental table).

RESPONSE

The journal requires that findings be replicated as a condition of publication. Furthermore, since initial sampling is within county subdivisions performed at randomized locations, replication of each scenario is essential to determine if the results are reproducible. When replicates did not provide consistent levels of accuracy relative to the HPAC plume, we demonstrated that both increased sampling densities and the more stringent overlap between the plumes improved the performance of the procedure. 

The information requested by the reviewer is quite lengthy and is not suitable for presentation as a Supplemental Table. As indicated above, locations and doses for each sample of every replicate of every scenario have been incorporated in the Zenodo archive along with the other data that we generated. 

Figure 2A: The numbers shown for Samples at first and final iteration (assuming >0Gy), BCD and RMSD for final vs HPAC plume do not agree with values provided in Table 1 and S1A. For B (Albany), only the sample number in first iteration (12) agrees with table S1A and Table 1.

RESPONSE

We investigated this issue. The New York (Urban) scenario in Figure 2 was derived before we began to prepare this manuscript. This replicate had been reported at the International Congress of Radiation Research in fall 2019, but was not previously described in the manuscript, which was based on an independent analysis. Details about the replicate in Figure 2 have now been added to both Table 1 and Supplementary Table S1A (and designated replicate #4), and the figure legend has been updated.

The Albany NY example present in Figure 2B is indeed Albany replicate #1 in Tables 1 and S1A. The BCD and RMSD were simply incorrectly transcribed for the Albany NY replicate from the original source analysis presented in S1A. We have corrected these statistics in Figure 2B. Also, as we indicated in the previous version of the manuscript, the “Number of Samples” column in Table 1 was updated to correct for duplicate locations computed from different densification steps for the same replicate. This issue persisted in Figure 2 but has been corrected.

In summary, this paper has the most value if it is clear to the naïve reader what was done and how the strategy might be used to advantage. And for a reader knowledgeable in the field who wants to examine the process in detail, there must be sufficient accurate and consistent data to allow that.

RESPONSE

We have more than adequately demonstrated the advantages of the proposed approach for deriving a dosimetry map from sparse sampling of the area impacted by a radiation plume. We also provide a detailed online step-by-step protocol requested by the journal, which the reviewer has failed to acknowledge. Following this protocol will allow a reader knowledgeable in the field to examine the process in detail and reconstruct our results.

The reviewer’s goal of testing the proposed method in real world scenarios is unrealistic. Our results can only be as accurate as the underlying population density data used to guide sampling. As stated previously, publicly available US census data is of lower geographic resolution than would be desirable for accurate sampling. Furthermore, the HPAC software that was available to us is not the current version provided by DTRA to US Government agencies. This software lacks many features, for example, it was not capable of incorporating the radiation shielding effects of buildings and infrastructure in urban environments, an issue that has been raised by this reviewer previously. These resources would be necessary in order to examine the process in detail at a sufficient accuracy and consistency to satisfy this reviewer. Nevertheless, the results we have achieved without these resources convincingly demonstrate that the procedures we developed are useful, because they closely approximate the HPAC plumes for all of the simulated scenarios that we analyzed.

---

## [Decision Letter · Decision Letter 4]

7 Apr 2020

Meeting radiation dosimetry capacity requirements of population-scale exposures by geostatistical sampling

PONE-D-19-22325R4

Dear Dr.Rogan:

We are pleased to inform you that your manuscript has been judged scientifically suitable for publication and will be formally accepted for publication once it complies with all outstanding technical requirements.

With kind regards,

Gayle E. Woloschak, PhD

Section Editor

PLOS ONE

Additional Editor Comments (optional):

Thank you for addressing the concerns of the reviewers.

Reviewers' comments:

Reviewer's Responses to Questions

**Comments to the Author**

1. If the authors have adequately addressed your comments raised in a previous round of review and you feel that this manuscript is now acceptable for publication, you may indicate that here to bypass the “Comments to the Author” section, enter your conflict of interest statement in the “Confidential to Editor” section, and submit your "Accept" recommendation.

Reviewer #1: All comments have been addressed

2. Is the manuscript technically sound, and do the data support the conclusions?

Reviewer #1: Yes

3. Has the statistical analysis been performed appropriately and rigorously? 

Reviewer #1: Yes

4. Have the authors made all data underlying the findings in their manuscript fully available?

Reviewer #1: Yes

5. Is the manuscript presented in an intelligible fashion and written in standard English?

Reviewer #1: Yes

6. Review Comments to the Author

Reviewer #1: (No Response)

7. PLOS authors have the option to publish the peer review history of their article (what does this mean?). If published, this will include your full peer review and any attached files.

Reviewer #1: No

---

## [Editor Report · Acceptance letter]

13 Apr 2020

PONE-D-19-22325R4 

Meeting radiation dosimetry capacity requirements of population-scale exposures by geostatistical sampling 

Dear Dr. Rogan:

I am pleased to inform you that your manuscript has been deemed suitable for publication in PLOS ONE. Congratulations! Your manuscript is now with our production department. 

With kind regards,

on behalf of

Dr. Gayle E. Woloschak 

Section Editor

PLOS ONE